# A General Identification Algorithm For Data Fusion Problems Under Systematic Selection

**Jaron J. R. Lee**[1]  **AmirEmad Ghassami**[2]  **Ilya Shpitser**[1]

[1]Department of Computer Science, Johns Hopkins University, Baltimore, Maryland, USA
[2]Department of Mathematics and Statistics, Boston University, Boston, Massachusetts, USA,

## Abstract

Identification of causal effects can be hampered by confounding, selection bias, and other complications. Data fusion is one approach to addressing these difficulties, through the inclusion of auxiliary data on the population of interest. Such data may measure a different set of variables, or be obtained under different experimental or observational conditions than the primary dataset. In particular, selection of experimental units into different datasets may be systematic; similar difficulties are encountered in missing data problems. However, existing methods for combining datasets either do not consider this issue, or assume simple selection mechanisms. In this paper, we propose a general approach, based on graphical causal models, for causal inference from data on the same population that is obtained under different experimental conditions. Our framework allows both arbitrary unobserved confounding, and arbitrary selection processes into different experimental regimes in our data. We describe how systematic selection processes may be organized into a hierarchy similar to censoring processes in missing data: selected completely at random, selected at random, and selected not at random. Finally, we provide a novel general identification algorithm for interventional distributions in this setting.

## 1 INTRODUCTION

Understanding causality is important for actionable insights. Traditionally, causality has been inferred through the use of randomized experiments, where an experimenter randomly assigns a variable $A$ to values corresponding to treatment or control, and measures the *causal effect* of these assignments on an outcome $Y$. However, such randomized experiments can be expensive, ethically fraught, or otherwise not possible to implement.

Given access to observational data, cause-effect relationships may be quantified via causal effects, which aim to predict what would have happened had a randomized experiment been (hypothetically) performed. A fundamental problem in causal inference is to estimate causal effects given access only to observational data and a causal model.

Causal effects may only be consistently estimated from observed data if they are *identified*, that is if they can be uniquely expressed as a function of observed data distribution under the assumptions encoded by a causal model. Sound and complete identification algorithms for causal effects have been developed using the formalism of graphical causal models [Shpitser and Pearl, 2006, Huang and Valtorta, 2008].

If the causal effect of interest is not identified, more assumptions may be imposed on the causal model, or informative conclusions about the causal effect may be obtained by deriving bounds. Alternatively, the primary dataset may be augmented with one or more informative secondary datasets, in what is termed *data fusion*. In this approach, an analyst has access to multiple datasets on the same population. These datasets may represent observational data on variables of interest, or represent results of randomized experiments, potentially ones that randomize treatments other than the treatments of primary interest. The task is to check if the desired causal effect can be computed from this collection of datasets. Sound and complete algorithms have been developed for this problem under the assumption that the collection of dataset is given [Lee et al., 2022, Kivva et al., 2022].

However, it is possible (indeed likely) that units are assigned to different datasets systematically, which is a possibility not considered by previous approaches. Consider the problem introduced by Athey et al. [2020], and further explored by Ghassami et al. [2022], where the goal is to estimate the effect of class size on student test scores in New York, using

two datasets. The first dataset is collected from the public school system, in which class size depends on observed and unobserved covariates that create confounding for the relationship between class size and test scores. The second dataset is from a randomized experiment studying the effect of class size conducted in a different population at a different point in time. As is typical in data fusion, the desired causal effect is not identified in either of datasets separately, and indeed the datasets measure differing sets of variables. However, the authors note that there are significantly different covariate distributions between the two. In other words, there is systematic selection determining which dataset a particular student ends up in. Therefore, appropriate adjustments are needed before causal conclusions obtained from the two subpopulation can be compared – or combined into a single conclusion on the overall population.

Our contributions are as follows. We define a novel graphical causal model for representing such data fusion problems, where the selection mechanism is represented as a random variable that potentially depends on other variables in the problem in complex ways. We use this representation to show that systematic selection exhibits a hierarchy similar to the missing data hierarchy, where selection could occur complete at random (SCAR), at random (SAR), or not at random (SNAR). Next, we show that applying a sound and complete algorithm that corrects for systematic selection following by a sound and complete algorithm that corrects for confounding cannot be complete in settings where both difficulties occur together. Finally, we propose a general algorithm that aims to correct for systematic selection and confounding at once.

## 2   BACKGROUND AND NOTATION

We use upper case Roman letters to denote random variables, e.g., $A$ and vector notation for sets thereof, e.g., $\vec{A}$. Values are lowercase letters, e.g., $a$, and sets of values are vectored lower case letters, e.g., $\vec{a}$. For a random variable $Z$, its domain is denoted as $\mathfrak{X}_Z$, and domains of sets $\vec{Z}$ as $\mathfrak{X}_{\vec{Z}}$. We denote the positive part of the domain (i.e. where $p(Z) > 0$) as $\mathfrak{X}_{\vec{Z}}^+$. Given a subset $\vec{A} \subseteq \vec{B}$ of variables, and values $\vec{b}$ of $\vec{B}$, we denote by $\vec{b}_{\vec{A}}$ the subset of values of $\vec{b}$ pertaining to variables in $\vec{A}$.

We use the framework of graphical causal modeling in this paper. Specifically, we will consider acyclic directed mixed graphs (ADMGs) which contain directed ($\rightarrow$) and bidirected ($\leftrightarrow$) edges and no directed cycles, and a special case of AD-MGs – directed acyclic graphs (DAGs) which contain only directed edges ($\rightarrow$). We employ the standard genealogical definitions for parents, ancestors, descendants, and districts of a variable $X$, as: $\mathrm{pa}_{\mathcal{G}}(X) = \{Y \mid Y \rightarrow X\}$, $\mathrm{an}_{\mathcal{G}}(X) = \{Y \mid Y \rightarrow \ldots \rightarrow X\} \cup \{X\}$, $\mathrm{de}_{\mathcal{G}}(X) = \{Y \mid X \rightarrow \ldots \rightarrow Y\} \cup \{X\}$, $\mathrm{dis}_{\mathcal{G}}(X) = \{Y \mid X \leftrightarrow \ldots \leftrightarrow Y\} \cup \{X\}$. In ad-

dition, we will define $\mathrm{nd}_{\mathcal{G}}(X)$ as the set of non-descendants of $X$, which is all vertices other than those in $\mathrm{de}_{\mathcal{G}}(X)$. These definitions apply disjunctively over sets - e.g., for set $\vec{Z}$, $\mathrm{pa}_{\mathcal{G}}(\vec{Z}) = \cup_{Z \in \vec{Z}} \mathrm{pa}_{\mathcal{G}}(Z)$. Strict versions of these definitions exclude variables in the argument, and are denoted with prepended $s$ - e.g., for set $\vec{Z}$, $\mathrm{spa}_{\mathcal{G}}(\vec{Z}) = \mathrm{pa}_{\mathcal{G}}(\vec{Z}) \setminus \vec{Z}$. We denote districts of a graph as $\mathcal{D}(\mathcal{G})$. Districts form a partition of vertices in a graph. Given a graph $\mathcal{G}$ with vertex set $\vec{V}$ and $\vec{Z} \subseteq \vec{V}$, define the induced subgraph $\mathcal{G}_{\vec{Z}}$ of $\mathcal{G}$ to be the graph containing vertices $\vec{Z}$ and edges in $\mathcal{G}$ only among elements in $\vec{Z}$.

We will consider structural causal models (SCMs), which are associated with DAGs. Given a DAG $\mathcal{G}$ with vertices $\vec{V}$ representing observed variables, we assume each variable $V \in \vec{V}$ is determined via an invariant mechanism called a structural equation: $f_V : \mathfrak{X}_{\mathrm{pa}_{\mathcal{G}}(V) \cup \{\epsilon_V\}} \mapsto \mathfrak{X}_V$, where $\epsilon_V$ is an exogenous unobserved random variable associated with $V$ representing the random noise in the system. We will assume that all $\epsilon_V$ are mutually independent. Some authors denote such an SCM as a non-parametric structural equation model with independent errors (NPSEM-IE) [Richardson and Robins, 2013b].

A causal model encodes responses to variables to the intervention operation, where structural equations of a set of variables $\vec{A}$ are replaced by constants $\vec{a}$. This operation is denoted by $\mathrm{do}(\vec{a})$ in Pearl [2009]. A random variable response of variable $Y$ to an intervention $\mathrm{do}(\vec{a})$ may also be written as a *potential outcome* $Y(\vec{a})$. Potential outcomes encode causal relationships in the sense that they allow representation of outcomes in hypothetical randomized controlled trials. For example, the average causal effect $\mathbb{E}[Y(a) - Y(a')]$ represents the difference in outcome response, on the mean difference scale, of two experimental groups, where a treatment $A$ is set to an active ($a$) or control ($a'$) value.

Since potential outcomes represent hypothetical changes in a causal system, responses to hypothetical interventions are not always available. An important task in causal inference is *identification,* ensuring that interventional distributions $p(\vec{Y}(\vec{a})) = p(\vec{Y}|\mathrm{do}(\vec{a}))$ are functions of the available distributions (classically, the observed data distribution $p(\vec{V})$).

It is well known that if all variables $\vec{V}$ in an SCM with independent errors are observed, every interventional distribution $p(\vec{V} \setminus \vec{A}|\mathrm{do}(\vec{a}))$ is identified via the truncated DAG factorization known as the *g-formula* [Robins, 1986]:

$$p(\vec{V} \setminus \vec{A} \mid \mathrm{do}(\vec{a})) = \prod_{V \in \vec{V} \setminus \vec{A}} p(V \mid \mathrm{pa}_{\mathcal{G}}(V))|_{\vec{a}_{\mathrm{pa}_{\mathcal{G}}(V) \cap \vec{A}}}.$$

A simple version of the g-formula is the adjustment formula, which yields the average causal effect of the treatment $A$ on the outcome $Y$ if all confounders of $A$ and $Y$ are observed as a vector $\vec{C}$: $\mathbb{E}[\mathbb{E}[Y|a, \vec{C}] - \mathbb{E}[Y|a', \vec{C}]]$. Note that the g-formula with the empty $\vec{A}$ also holds and implies that the observed data distribution $p(\vec{V})$ may be written as

$\prod_{V \in \vec{V}} p(V \mid \mathrm{pa}_{\mathcal{G}}(V))$, and thus is Markov with respect to the DAG $\mathcal{G}$ [Pearl, 1988, Lauritzen, 1996].

# 3  THE ID ALGORITHM

If some variables in the model are unobserved, identification of interventional distributions becomes considerably more complicated, with some distributions not being identified at all, and distributions that are identified being potentially more complex functionals of the observed data distribution than the g-formula. General identification algorithms given the observed marginal distribution $p(\vec{V})$ derived from a hidden variable causal model represented by a DAG $\mathcal{G}(\vec{V} \cup \vec{H})$, where $\vec{H}$ represent hidden variables have been characterized via the ID algorithm [Tian and Pearl, 2002, Shpitser and Pearl, 2006]. The ID algorithm takes as input an ADMG $\mathcal{G}(\vec{V})$ derived from $\mathcal{G}(\vec{V} \cup \vec{H})$ via the latent projection operation [Verma and Pearl, 1990], the observed data distribution $p(\vec{V})$, and disjoint variable sets $\vec{A}, \vec{Y}$ corresponding to the interventional distribution $p(\vec{Y} \mid \mathrm{do}(\vec{a}))$ of interest. The ID algorithm outputs either the identifying functional for $p(\vec{Y} \mid \mathrm{do}(\vec{a}))$ in terms of $p(\vec{V})$, or the token "not identified." It is known that the ID algorithm is both sound (outputs correct identifying functionals in all cases) and complete (whenever it outputs "not identified," the corresponding distribution is indeed not a function of $p(\vec{V})$ in the model) [Huang and Valtorta, 2006, Shpitser and Pearl, 2006].

Just as the g-formula is a one line formula representing a modified DAG factorization, the ID algorithm may be formulated as a one line formula representing a modified nested Markov factorization of a latent projection ADMG [Richardson et al., 2023]. We now briefly review the ID algorithm formulated in this way in terms of Markov kernels, and the fixing operator.

A Markov kernel $q_{\vec{V}}(\vec{V} \mid \vec{W})$ is a nonnegative function that marginalizes to 1 over $\vec{V}$ for values of $\mathfrak{X}_{\vec{W}}$, and may be viewed as a generalization of a conditional distribution, but is not necessarily constructed by applying a conditioning operation to a joint distribution. For example, the kernel $q_Y(Y \mid a) \equiv \sum_{\vec{C}} p(Y \mid a, \vec{C}) p(\vec{C})$ which appears in the adjustment formula is not, in general, equal to $p(Y \mid a)$.

Given a Markov kernel, additional kernels may be constructed by the conditioning and marginalization operators, which are defined in the natural way:

$$q_{\vec{V}}(\vec{B} \mid \vec{W}) \equiv \sum_{\vec{V} \setminus \vec{B}} q_{\vec{V}}(\vec{V} \mid \vec{W}); q_{\vec{V}}(\vec{V} \setminus \vec{B} \mid \vec{B} \cup \vec{W}) = \frac{q_{\vec{V}}(\vec{V} \mid \vec{W})}{q_{\vec{V}}(\vec{B} \mid \vec{W})}.$$

The fixing operator [Richardson et al., 2023] is an operator applied to graphs and kernels that "removes" vertices and random variables by rendering them "fixed". A relevant generalization of an ADMG called a conditional ADMG (CADMG) $\mathcal{G}(\vec{V}, \vec{W})$ contains two types of vertices: random

(denoted by $\vec{V}$), and fixed (denoted by $\vec{W}$). Fixed vertices cannot have any edges with an arrowhead into them, and will be displayed as squares in graphs. Note that an ADMG is a CADMG where $\vec{W}$ is empty. Kernels $q_{\vec{V}}(\vec{V} \mid \vec{W} = \vec{w})$ and CADMGs $\mathcal{G}(\vec{V}, \vec{W})$ will represent interventional distributions $p(\vec{V} \mid \mathrm{do}(\vec{w}))$, and their Markov structure, respectively. *Mutilated graphs* used in Pearl [2009] to describe interventional contexts may be viewed as CADMGs, provided intervened on variables are distinguished from variables that remain random.

For a CADMG $\mathcal{G}(\vec{V}, \vec{W})$, a vertex $V \in \vec{V}$ is fixable if there is no other vertex that is both a descendant and in the same district in $\mathcal{G}$, that is if $\mathrm{dis}_{\mathcal{G}}(V) \cap \mathrm{de}_{\mathcal{G}}(V) = \{V\}$. If $V$ is fixable we define a new CADMG $\mathcal{G}(\vec{V} \setminus \{V\}, \vec{W} \cup \{V\}) \equiv \phi_V(\mathcal{G}(\vec{V}, \vec{W}))$ by means of a fixing operator $\phi_V$ which renders $V$ a fixed vertex, removes all edges in $\mathcal{G}(\vec{V}, \vec{W})$ with an arrowhead into $V$, and keeps all other vertices and edges unaltered.

Given a non-empty sequence of vertices $\pi$, we define $h(\pi)$ to be its first element, and $t(\pi)$ to be the subsequence of $\pi$ containing all elements after the first. A sequence $\pi$ of vertices in $\vec{V}$ is said to be valid (or fixable) in a CADMG $\mathcal{G}(\vec{V}, \vec{W})$ if either the sequence is empty, or $h(\pi)$ is fixable in $\mathcal{G}$ and $t(\pi)$ is fixable in $\phi_{h(\pi)}(\mathcal{G})$. Any two sequences on the same set of vertices $\vec{S} \subseteq \vec{V}$ fixable in $\mathcal{G}(\vec{V}, \vec{W})$ yield the same CADMG, allowing us to write $\phi_{\vec{S}}(\mathcal{G})$ to mean "obtain the CADMG after applying the fixing operator to $\vec{S}$ via any valid sequence". A set $\vec{R} \subseteq \vec{V}$ is said to be reachable in an ADMG $\mathcal{G}$ with vertices $\vec{V}$ if there exists a valid fixing sequence for $\vec{V} \setminus \vec{R}$. Given a set $\vec{R} \subseteq \vec{V}$ that is not reachable in $\mathcal{G}$, the unique smallest reachable superset of $\vec{R}$ that is reachable of $\mathcal{G}$ is called a *reachable closure of $\vec{R}$*, or simply closure of $\vec{R}$, and denoted by $\mathrm{cl}_{\mathcal{G}}(\vec{R})$.

Given a kernel $q_{\vec{V}}(\vec{V} \mid \vec{W})$ associated with a CADMG $\mathcal{G}(\vec{V}, \vec{W})$ where $V$ is fixable, define $\phi_V(q_{\vec{V}}; \mathcal{G})$ to be the kernel $q_{\vec{V} \setminus \{V\}}(\vec{V} \setminus \{V\} \mid \vec{W} \cup \{V\}) \equiv \frac{q_{\vec{V}}(\vec{V} \mid \vec{W})}{q_{\vec{V}}(V \mid \mathrm{nd}_{\mathcal{G}}(V), \vec{W})}$. Given a CADMG $\mathcal{G}(\vec{V}, \vec{W})$, kernel $q_{\vec{V}}(\vec{V} \mid \vec{W})$ and a sequence $\pi$ of vertices in $\vec{V}$ fixable in $\mathcal{G}$, we define $\phi_\pi(q_{\vec{V}}, \mathcal{G})$ as $q_{\vec{V}}$ if $\pi$ is the empty sequence, and as $\phi_{t(\pi)}(\phi_{h(\pi)}(q_{\vec{V}}; \mathcal{G}); \phi_{h(\pi)}(\mathcal{G}))$ otherwise. Given $p(\vec{V})$ which is a marginal distribution obtained from $p(\vec{V} \cup \vec{H})$ which is Markov with respect to a DAG $\mathcal{G}(\vec{V} \cup \vec{H})$, and corresponding latent projection ADMG $\mathcal{G}(\vec{V})$ of $\mathcal{G}(\vec{V} \cup \vec{H})$, and any two sequences $\pi_1, \pi_2$ on $\vec{S} \subseteq \vec{V}$ valid in $\mathcal{G}$, $\phi_{\pi_1}(q_{\vec{V}}; \mathcal{G}) = \phi_{\pi_2}(q_{\vec{V}}; \mathcal{G})$. We thus denote the resulting kernel by $\phi_{\vec{S}}(q_{\vec{V}}; \mathcal{G})$.

The identification of interventional distributions $p(\vec{Y} \mid \mathrm{do}(\vec{a}))$ for any disjoint subsets $\vec{A}, \vec{Y}$ of $\vec{V}$ in a hidden variable causal model associated with a DAG $\mathcal{G}(\vec{V} \cup \vec{H})$ with a latent projection $\mathcal{G}(\vec{V})$ has been characterized as follows. Let $\vec{Y}^* = \mathrm{an}_{\mathcal{G}(\vec{V})_{\vec{V} \setminus \vec{A}}}(\vec{Y})$, and define $\mathcal{G}^*$ as $\mathcal{G}(\vec{V})_{\vec{Y}^*}$ Then

$p(\vec{Y}|\mathrm{do}(\vec{a}))$ is identified if and only if every element in $\mathcal{D}(\mathcal{G}(\vec{V})_{\vec{Y}*})$ is reachable in $\mathcal{G}(\vec{V})$. If so, the reformulation of the ID algorithm by Richardson et al. [2023] gives

$$p(\vec{Y}|\mathrm{do}(\vec{a})) = \sum_{\vec{Y}*\setminus\vec{Y}} \prod_{\vec{D}\in\mathcal{D}(\mathcal{G}(\vec{V})_{\vec{Y}*})} p(\vec{D}|\mathrm{do}(\mathrm{spa}_{\mathcal{G}(\vec{V})}(\vec{D}))) \quad (1)$$

$$= \sum_{\vec{Y}*\setminus\vec{Y}} \prod_{\vec{D}\in\mathcal{D}(\mathcal{G}(\vec{V})_{\vec{Y}*})} \phi_{\vec{V}\setminus\vec{D}}(p(\vec{V}); \mathcal{G}(\vec{V}))|_{\vec{A}=\vec{a}}.$$

## 4 THE GID ALGORITHM: A REVIEW

Consider an observational study $\mathcal{S}_1$ which aimed to assess antibiotic effectiveness for treating infections. In practice, patients are prone to non-compliance, where the full course is not taken as instructed. We represent this causal structure with antibiotic prescription $A$, compliance level $M$, and a longer term outcome variable $Y$ such as hospital 30-day readmission. Unobserved common causes $U_1, U_2, U_3$ create confounding for several study variables, but a pre-treatment proxy $C$ of $U_2, U_3$ is observed. The ID algorithm demonstrates that $p(Y(a))$ is not identified from data on observed variables in the model shown in Fig. 1a.

Now consider an experimental study $\mathcal{S}_2$, in which the investigator is able to control compliance level based on assigned treatment. This results in Fig. 1b, in which the edge $A \to M$ is kept, but the edge $U_2 \to M$ is removed, representing the situation in which compliance status only depends on the prescribed medication (and not, say, the patient's mood or appetite). It turns out that if both datasets are drawn from the same population, the causal effect $p(Y(a))$ may be obtained from the combined dataset as $\sum_{c,m,\tilde{a}} \mathbb{E}_2[Y \mid m, \tilde{a}] p_1(\tilde{a}) \sum_c p_1(m \mid a, c) p_1(c)$, where subscripts indicate distributions computed from datasets $\mathcal{S}_1$ or $\mathcal{S}_2$. Lee et al. [2019] provided a sound and complete graphical *gID algorithm* for the identification of such queries, in the special case where experimental datasets are formulated only using the do(.) operator, rather than more complex policy interventions like in the above example that may depend on other variables in the problem.

We now reformulate the gID algorithm using the fixing operator $\phi$ (see also Lee and Bareinboim [2020], Lee and Shpitser [2020]). Given the interventional distribution of interest $p(\vec{Y}(\vec{a}))$, and a latent projection ADMG $\mathcal{G}(\vec{V})$ representing a hidden variable causal model, let the available datasets be denoted $\{p_i(\vec{V} \mid \mathrm{do}(\vec{Z}_i))\}_{i=1}^K$ with corresponding CADMGs $\{\mathcal{G}_i(\vec{V} \setminus \vec{Z}_i, \vec{Z}_i)\}_{i=1}^K$. Note that these CADMGs need not have been obtained via the fixing operator. Then, if for each $\vec{D} \in \mathcal{D}(\mathcal{G}(\vec{V})_{\vec{Y}*})$ there exists $j_{\vec{D}} \in \{1, \ldots, K\}$ such that $\vec{D}$ is reachable in $\mathcal{G}_{j_{\vec{D}}}(\vec{V} \setminus \vec{Z}_{j_{\vec{D}}}, \vec{Z}_{j_{\vec{D}}})$, then output

$$p(\vec{D} \mid \mathrm{do}(\mathrm{spa}_{\mathcal{G}(\vec{V})}(\vec{D}))) = \phi_{(\vec{V}\setminus\vec{Z}_{j_{\vec{D}}})\setminus\vec{D}}(p_{j_{\vec{D}}}; \mathcal{G}_{j_{\vec{D}}}).$$

The causal effect is then obtained by the usual district factorization (1) as in the ID algorithm.

The primary limitation of this and related prior work is that the selection process is *under-specified*. In the case of gID, the selection process is not represented at all, as nothing is specified about how distributions $\{p_i(\vec{V} \mid \mathrm{do}(\vec{Z}_i))\}_{i=1}^K$ are related. In other related work such as transportability [Bareinboim and Pearl, 2012], selection bias [Bareinboim and Tian, 2015], or as a general representation of interventional and observational domains in causal inference [Dawid, 2021], the selectors enter the model but only as non-random indicators that index domains.

Since domain selectors are not treated as full random variables, the resulting models do not yield a single coherent data likelihood, which is an impediment to statistical inference. A more serious issue, however, is that by not modeling the selection process explicitly, it is not possible to represent *systematic selection*, which is often how units from a single superpopulation are assigned to different experimental and observational settings in practice.

To address these issues, we will represent the selection process as a random variable $S$. In this paper, this variable indexes intervention status of its children in the graph, but may also indicate changes in structural equations representing domain differences. Selectors may potentially share common (and potentially unobserved) parents with other variables, creating potential confounding.

## 5 CAUSAL MODELS FOR SELECTION

In this section, we describe how to augment SCMs with an additional selector random variable $S$ that governs whether certain variables that are its children in the causal graph are intervened on or keep their natural behavior.

**Definition 1** (Context Selected SCM). *Given an SCM with independent errors associated with a DAG $\mathcal{G}(\vec{V})$, a context selected SCM (CS-SCM) associated with a DAG $\bar{\mathcal{G}}(\vec{V}\cup\{S\})$, such that $\mathrm{ch}_{\bar{\mathcal{G}}}(S) \neq \emptyset$ and $\mathrm{pa}_{\bar{\mathcal{G}}}(S)$ is arbitrary, is defined as follows:*

- $S \equiv \{\langle S^e_{\mathrm{ch}_{\bar{\mathcal{G}}}(S)}, S^v_{\mathrm{ch}_{\bar{\mathcal{G}}}(S)}\rangle \mid S^e_{\mathrm{ch}_{\bar{\mathcal{G}}}(S)} \in \mathfrak{X}_{\mathrm{ch}_{\bar{\mathcal{G}}}(S)} \subseteq \{0,1\}^{|\mathrm{ch}_{\bar{\mathcal{G}}}(S)|}, S^v_{\mathrm{ch}_{\bar{\mathcal{G}}}(S)} \in \mathfrak{X}_{\mathrm{ch}_{\bar{\mathcal{G}}}(S)} \equiv \otimes_{V\in\mathrm{ch}_{\bar{\mathcal{G}}}(S)}\mathfrak{X}_V\}$, *and furthermore* $\mathfrak{X}^+_{\mathrm{ch}_{\bar{\mathcal{G}}}(S)} \equiv \otimes_{\mathrm{ch}_{\bar{\mathcal{G}}}(S)}\mathfrak{X}^+_V$.

- *Every $V \in \vec{V} \setminus \mathrm{ch}_{\bar{\mathcal{G}}}(S)$ maintains its structural equation $f_V(\mathrm{pa}_{\mathcal{G}}(V), \epsilon_V)$ from the original SCM.*

- *For every $V \in \vec{V} \cap \mathrm{ch}_{\bar{\mathcal{G}}}(S)$, the structural equation $\tilde{f}_V$ for $V$ in the CS-SCM is defined in terms of $S$ and the structural equation $f_V(\mathrm{pa}_{\mathcal{G}}(V), \epsilon_V)$ for $V$ in the original SCM as:*

$$V \leftarrow \tilde{f}_V(\mathrm{pa}_{\mathcal{G}}(V), S, \epsilon_V)$$

$$\tilde{f}_V(\mathrm{pa}_{\mathcal{G}}(V), S, \epsilon_V) \equiv \begin{cases} f_V(\mathrm{pa}_{\mathcal{G}}(V), \epsilon_V) & \text{if } S^e_V = 0 \\ S^v_V & \text{if } S^e_V = 1 \end{cases}$$

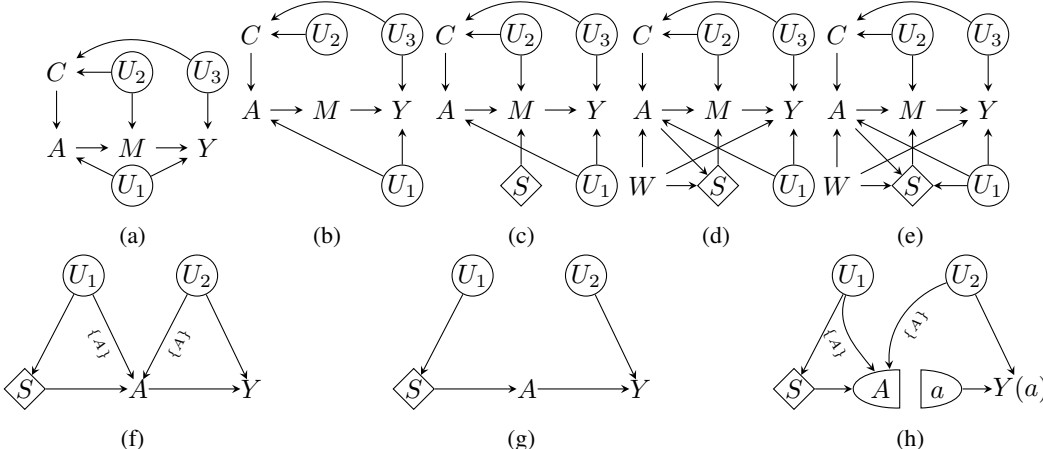

Figure 1: Example graphs illustrating systematic selection. Unobserved variables are denoted with a enclosing circle, while observed variables are not. The (observed) selection variable $S$ is denoted by a diamond. Half-circles represent the split-node operation in SWIGs [Richardson and Robins, 2013a].

- *For $S$, its structural equation is specified by a new equation*

$$S \leftarrow f_S(\mathrm{pa}_{\bar{\mathcal{G}}}(S), \epsilon_S)$$

*where $\mathrm{pa}_{\bar{\mathcal{G}}}(S)$ is chosen to avoid introducing cycles.*

In words, the CS-SCM is an SCM augmented with a selector variable $S$ consisting of intervention indicators $S_V^e$ and intervention values $S_V^v$ for every child $V$ of $S$. If $S_V^e = 1$, the child $V$ of $S$ is intervened on, and $S_V^v$ indicates the value of the intervention. If $V$ is not intervened on, $V$ acts as a usual function of its parents other than $S$ via its structural equation $f_V$ from the original SCM. To simplify notation, when discussing CS-SCMs, we will include $S$ in the set $\vec{V}$, and denote values of $S$ as a vector pair $\langle \bar{s}^e, \bar{s}^v \rangle$.

We note that the relationship of the random selector $S$ and its children we describe here is similar closely related to *context variables* in the joint causal modeling approach in Mooij et al. [2020]. Our approach is also related to the decision-theoretic framework in Dawid [2021] and selection diagrams applied to selection bias and transportability problems in Bareinboim and Tian [2015], Bareinboim and Pearl [2012], although these approaches do not treat their respective selectors as fully random variables in the causal model.

In subsequent developments we will make use of the following definition. Given a set of variables $\vec{D} \subseteq \vec{V}$ in a CS-SCM with associated DAG $\mathcal{G}$, we say a value $s = \langle \bar{s}^e, \bar{s}^v \rangle$ is *laidback for $\vec{D}$* if for each $V \in \vec{D}$, either $\bar{s}_V^e = 0$, or $V \notin \mathrm{ch}_{\mathcal{G}}(S)$. We say $s$ is *serious for $\vec{D}$* if it is not laidback for $\vec{D}$. We will also use the abuse of notation $s = \emptyset$ to denote any value set $\langle \bar{s}^e, \bar{s}^v \rangle$, where $\bar{s}_{\vec{V}}^e = \vec{0}$ for observed variables $\vec{V}$. Such value sets correspond to the observational context of an CS-SCM, the context where no interventions took place.

The CS-SCM exhibits *context-specific* independencies, whereby a variable is no longer a function of some others at particular values of a parent. Pensar et al. [2015] provide a succinct graphical representation of this information.

**Definition 2** (Labelled selection DAG). *Let $\mathcal{G}(\vec{V})$ denote a DAG associated with a CS-SCM with edges $\vec{E}$. For each edge $E = (A \rightarrow B) \in \vec{E}$ such that $S \in \mathrm{pa}_{\mathcal{G}}(B)$ and $S \neq A$, we attach a label $L_E = \{B\}$. Then, $\mathcal{G}^{[]}(\vec{V})$ with edge labels $\vec{L} = \cup_E L_E$ denotes a labelled selection DAG (LS-DAG) associated with the CS-SCM.*

Given an LS-DAG $\mathcal{G}^{[]}$ and a value $s$ of $S$, we define the *context selected* DAG $\mathcal{G}^{[s]}$ to be an edge subgraph of $\mathcal{G}^{[]}$ where any edge with a label $\{V\}$ is removed if $\bar{s}_V^e = 1$, and removes all other edge labels. Note that the graph $\mathcal{G}^{[\emptyset]}$ removes all edge labels but no edges. Fig. 1f is an example of an LS-DAG $\mathcal{G}^{[]}$, while Fig. 1g is an example of $\mathcal{G}^{[s]}$ for a value $s$ such that $\bar{s}_A^e = 1$.

While we developed the CS-SCM in this section, we note that such models are cross-world models due to the independence of error terms in the NPSEM-IE. Since we do not rely on cross-world independencies elsewhere in this paper, this section can in principle be reformulated using single-world models such as the finest fully randomized causally interpretable structured tree graph, or FFRCISTG [Robins, 1986]. The principle difference between these models is that FFRCISTG assumptions can in theory be empirically verified under hypothetical randomized experiments where any subset of variables can be intervened upon, whereas this is not true in an NPSEM-IE as well as the CS-SCM that we defined. Shpitser et al. [2021] provide further details on this distinction.

## 5.1 SWIGS AND CONTEXT SELECTED SWIGS

Given a DAG $\mathcal{G}(\vec{V})$ representing an SCM, and a set of values $\vec{a}$ of treatments $\vec{A}$, a single world intervention graph (SWIG) [Richardson and Robins, 2013b] $\mathcal{G}(\vec{V}(\vec{a})) = \mathcal{G}(\vec{a})$ is a graph obtained from $\mathcal{G}$ by creating a "random" version $A$ and a "fixed" version $a$ of every $A \in \vec{A}$ vertex in $\mathcal{G}$, with every random version $A$ inheriting all edges with arrowheads into $A$ in $\mathcal{G}$, and every fixed version $a$ inheriting all outgoing edges from $A$ in $\mathcal{G}$. In addition every vertex $V$ in $\mathcal{G}(\vec{a})$ is relabelled as $V(\vec{a})$ to signify that these vertices represent counterfactual random variables.

A SWIG $\mathcal{G}(\vec{a})$ represents Markov structure of an interventional distribution $p(\vec{V}(\vec{a}))$ obtained from an SCM with a graph $\mathcal{G}$ via the d-separation criterion [Richardson and Robins, 2013b]. Standard genealogic relations generalize readily to SWIGs.

We consider a special case of SWIGs applicable to our setting. Given a CS-SCM associated with an LS-DAG $\mathcal{G}^{[]}$ with vertices $\vec{V}$, if $S \notin \vec{A}$, we represent $p(\vec{V}(\vec{a}))$ via a *labelled selection SWIG (LS-SWIG)* $\mathcal{G}^{[]}(\vec{a})$, which is obtained by employing the standard SWIG construction while keeping the labels in $\bar{\mathcal{G}}^{[]}$.

If $S \in \vec{A}$, let $s$ be the value of $S$ in $\vec{a}$. Then we define a context-specific SWIG $\bar{\mathcal{G}}^{[s]}(\vec{a})$ as follows. Given an LS-SWIG $\bar{\mathcal{G}}^{[]}(\vec{a})$, we remove any random vertex in $\mathrm{ch}_{\bar{\mathcal{G}}(\vec{a})}(s)$ that $s$ is serious for, and all edges adjacent to such vertices. This operation represents the fact that such a vertex corresponds to a constant.

Despite removal of certain vertices and edges, context-specific SWIGs correctly represent independences in interventional distributions obtained from CS-SCMs due to the following result.

**Theorem 1.** *Given a CS-SCM associated with $\mathcal{G}^{[]}(\vec{V})$, and any $\vec{A} \subseteq \vec{V}$ such that $S \in \vec{A}$ (including $\vec{A} = \{S\}$), any d-separation statement in $\mathcal{G}^{[s]}(\vec{a})$, for $s$ consistent with $\vec{a}$, implies a conditional independence statement in $p(\vec{V}(\vec{a}))$.*

## 5.2 THE SELECTION HIERARCHY AND CONTEXT SELECTED G-FORMULA

Treating the selector $S$ as a part of the model allows us to represent systematic selection via a hierarchy similar to the missing data hierarchy [Rubin, 1976], with selected completely at random (SCAR), selected at random (SAR), and selected not at random (SNAR) models. In particular, we can recast the earlier antibiotic example as a SCAR model, by assuming that assignment into the different studies $\mathcal{S}_1, \mathcal{S}_2$ is random and that $S$ has no causes Fig. 1c. One can view the SCAR model as the generalization "closest in spirit" to the original gID formulation that admits a coherent observed data likelihood that includes both observational and interventional contexts.

SCAR models, like MCAR models in missing data, are often unrealistic, as we expect selection into different domains to be systematic. In our example, if the selection mechanism into either the observational group or the experimental study is influenced by observed characteristics $W$, such as the patient's age, as well as the treatment assignment $A$, the result is a SAR model shown in (Fig. 1d). If the patients are also selected based on unobserved characteristics that also influence patient outcomes, such as a doctor's intuition about a particular case $U_1$, the result is a SNAR model shown in (Fig. 1e).

Since $S$ is a part of the model, representing situations where only some interventions are available to the analyst entails imposing restrictions on support of $S$. Thus, we allow only a subset of $\mathfrak{X}_S$, termed $\mathfrak{X}_S^+$, to have support. For example, if $S$ has children $A_1$ and $A_2$, we may allow $\mathfrak{X}_{\{S_{A_1}^e, S_{A_2}^e\}}$ to have support on the set $\{\{0, 0\}, \{0, 1\}, \{1, 0\}\}$. In other words, $S$ allows no variables to be intervened on, or either only $A_1$ or $A_2$ to be intervened on, but not both $A_1$ and $A_2$. Prior work represented this by explicitly providing a set of distributions as inputs to the algorithm [Lee et al., 2019].

Queries corresponding to interventional distributions $p(\vec{Y}(\vec{a}))$ in an SCM must be modified in a CS-SCM to take the special nature of $S$ into account. In particular, the analogue of the query $p(\vec{Y}(\vec{a}))$ in the original SCM corresponds to $p(\vec{Y}(\vec{a}, S = \emptyset))$, which reads "the distribution of outcomes $\vec{Y}$, when the context of the CS-SCM is set to the observational value, and the variable $\vec{A}$ is set to $\vec{a}$". Intuitively, this excludes contexts where variables such as $\vec{Y}$ are intervened, and whose intervened distributions are not of scientific interest. Note that this query potentially entails a positivity violation in the sense that no positive support may exist in the observed data distribution for the situation where $\vec{A} = \vec{a}$ and $S = \emptyset$. This occurs, in particular if elements of $\vec{A}$ are among children of $S$. While this may potentially prevent identification, restrictions on the CS-SCM may allow identification to be obtained in some cases. A close analog of this phenomenon arises in the interventionist formulations of mediation analysis [Robins and Richardson, 2010, Robins et al., 2023].

If all variables in an CS-SCM are observed, we have the following result for the query $p(\vec{Y}(\vec{a}, S = \emptyset))$ that directly generalizes the g-formula.

**Theorem 2** (Context Selected g-formula). *Fix a fully observed CS-SCM corresponding to an LS-DAG $\mathcal{G}^{[]}$ with a vertex set $\vec{V}$, and disjoint subsets $\vec{A}, \vec{Y}$ of $\vec{V}$. Let $\vec{Y}^* = \mathrm{an}_{\mathcal{G}^{[]}(\vec{a}, \emptyset)}(\vec{Y})$ Then $p(\vec{Y}(\vec{a}, S = \emptyset))$ is identified if and only if for every element $V \in \vec{Y}^*$ there exists a value $s_V \in \mathfrak{X}_S^+$ laidback for $V$ (i.e. $s_V^e = 0$). If so, we have:*

$$p(\vec{Y}(\vec{a}, S = \emptyset)) = \sum_{\vec{Y}^* \setminus \vec{Y}} \prod_{V \in \vec{Y}^*} p(V \mid \mathrm{pa}_{\mathcal{G}}(V))|_{\vec{a}_{\vec{A} \cap \mathrm{pa}_{\mathcal{G}}(V)}, S_V^e = 0}.$$

(2)

The above g-formula takes into consideration the requirement that $S = \emptyset$, which ensures that the causal effect is computed in the observational context only.

Note that the query may not be identified even under full observability, if available contexts for $S$ are not laidback for elements in $\vec{Y}^*$. Nevertheless, the above result allows identifiability in situations corresponding to SCAR or SAR. While incorporating the selection process as an explicit part of the causal model allows us to explicitly represent complex types of systematic selection, it also (unsurprisingly) creates difficulties with identification of causal effects in models with hidden variables that yield systematic selection more complicated than SCAR or SAR.

## 5.3 LATENT PROJECTIONS IN CS-SCMS

In order to formulate a general identification algorithm for CS-SCMs with hidden variables, we first generalize latent projections and the fixing operator to CS-SCMs.

A *labelled selection acyclic directed mixed multigraph (LS-ADMMG)* is a multigraph with directed and bidirected edges, no directed cycles, and the property that any pair of edges of the same type connecting the same vertex pair $A, B$ must have different labels. Given an LS-DAG $\mathcal{G}^{[]}(\vec{V} \cup \vec{H})$, where $S \in \vec{V}$, and where labels may exist on any edge in this LS-DAG, define a latent projection $\mathcal{G}^{[]}(\vec{V})$ to be an LS-ADMMG with vertices $\vec{V}$, where for each directed path from $A \in \vec{V}$ to $B \in \vec{V}$ in $\mathcal{G}^{[]}(\vec{V} \cup \vec{H})$ where all intermediate elements are in $\vec{H}$, a directed edge labeled by a union of labels for every edge on the path is added to $\mathcal{G}(\vec{V})$. Similarly, for each marginally d-connecting path from $A$ to $B$ in $\bar{\mathcal{G}}^{[]}(\vec{V} \cup \vec{H})$, where the first edge is into $A$ and the last into $B$ and where all intermediate elements are in $\vec{H}$, add to $\mathcal{G}(\vec{V})$ a bidirected edge labelled by a union of labels for every edge on this path. Note that the result is a multigraph since the same pair may be connected by the same edge type with multiple labels. A similar definition yields a latent projection $\mathcal{G}^{[]}(\vec{V}(\vec{a}))$ of a labelled hidden variable SWIG $\mathcal{G}^{[]}(\vec{V}(\vec{a}) \cup \vec{H}(\vec{a}))$. An example illustrating why labelled multigraphs are necessary to represent latent projections of LS-DAGs in general is found in the Appendix.

Similarly, we define a labelled selection conditional AD-MMG (LS-CADMMG) as an LS-ADMMG with random and fixed vertices, such that fixed vertices cannot have edges with arrowheads into them.

Given an LS-CADMMG $\mathcal{G}^{[]}(\vec{V}, \vec{W})$ where $S \in \vec{W}$, the *context selected graph* $\mathcal{G}^{[s]}(\vec{V}, \vec{W})$ is defined by removing every edge such that $s$ is serious for any vertex in that edge's label, removing labels for all other edges, and removing every unlabelled duplicate edge of the same type connecting every pair of vertices. Note that this construction always yields a CADMG. Note also that if $s = \emptyset$, all parents and

siblings of $\mathcal{G}$ are preserved in $\mathcal{G}^{[s]} = \mathcal{G}^{[\emptyset]}$, but the resulting object is no longer a multigraph.

The fixing operator and genealogic relations generalize in a straightforward way to multigraphs we consider. In particular, multiple labelled edges of the same type connecting vertices $A$ and $B$ are treated as a single edge of that type, with labels ignored. If a graph index for a genealogic set is omitted, it is understood to be $\mathcal{G}^{[\emptyset]}$.

## 5.4 TOWARDS IDENTIFICATION UNDER SNAR

A seemingly reasonable approach for obtaining identification of interventional distributions $p(\vec{Y}(\vec{a}, S = \emptyset))$ given a hidden variable CS-SCM represented by an LS-ADMMG $\mathcal{G}^{[]}(\vec{V})$ is to first address systematic selection by identifying $q_{\vec{V}}(\vec{V}) \equiv p(\vec{V}|do(S = \emptyset))$, corresponding to the SWIG $\mathcal{G}^{[\emptyset]}(\emptyset)$, and then invoke the ID algorithm on this CADMG, the distribution $p(\vec{V})$ and the query $p(\vec{Y}(\vec{a}, S = \emptyset))$. This strategy clearly yields a sound algorithm. In fact, we can show that despite the extra context-specific independencies implied by an CS-SCM, we have the following result.

**Theorem 3.** *Given a hidden variable CS-SCM represented by a LS-ADMMG $\mathcal{G}^{[]}(\vec{V})$, the ID algorithm with causal query $p(\vec{V}|do(S = \emptyset))$, data distribution $p(\vec{V})$, and ADMG $\mathcal{G}^{[\emptyset]}$ is sound and complete.*

Despite this, the above sequential strategy does not yield a complete algorithm for systematic selection for the more general causal query $p(\vec{Y}(\vec{a}, S = \emptyset))$. To see why, consider the following simple example, illustrated by the hidden variable LS-DAG $\bar{\mathcal{G}}^{[]}$ shown in Fig. 1f, where we are interested in $p(Y(a, s = \emptyset))$. Completeness of the ID algorithm implies that this interventional distribution is not identified under standard SCM semantics corresponding to this graph. Theorem 3 above also implies the distribution $p(Y, A \mid do(S = \emptyset))$ is not identified.

However, identification is obtained in an CS-SCM due to the following simple derivation:

$$p(Y(a, S = \emptyset)) = p(Y(a)) = p(Y(a)|S = (s_A^e = 1, s_A^v = a))$$
$$= p(Y(a) \mid A = a, S = (s_A^e = 1, s_A^v = a))$$
$$= p(Y \mid A = a, S = (s_A^e = 1, s_A^v = a)).$$

Here the first equality follows by the exclusion restrictions in this model (or by rule 3 of the potential outcomes calculus [Malinsky et al., 2019]). The second equality follows since $Y(a) \perp\!\!\!\perp S$ in this model, which may be verified by the context selected SWIG shown in 1h. The third equality follows by definition of the CS-SCM, and the final equality by consistency.

This derivation is explained by noting that $S$ acts as a *perfect instrument* for the effect of $A$ on $Y$. Specifically $S$ only influences $Y$ through $A$, and $S$ is independent of any

confounders for $A$ and $Y$. In addition, unlike standard instruments, $S$ completely determines the value of $A$, thereby eliminating any influence of the confounder $U_2$ on $A$. In light of examples like above, we formulate a general identification algorithm that is able to handle both systematic selection and unobserved confounding together.

# 6 AN IDENTIFICATION ALGORITHM FOR SYSTEMATIC SELECTION

Here, we present a general identification algorithm, shown as Algorithms 1 and 2, for the query $p(\vec{Y}(\vec{a}, S = \emptyset))$ in a hidden variable CS-SCM represented by a latent projection multigraph $\mathcal{G}^{[]}(\vec{V})$, where $S \in \vec{V}$. The algorithm proceeds from the usual factorization used by both the ID and gID algorithms:

$$p(\vec{Y}(\vec{a}, S = \emptyset)) = \sum_{\vec{Y}^* \backslash \vec{Y}} \prod_{\vec{D}^* \in \mathcal{D}(\mathcal{G}^{[]}(\vec{a}, \emptyset))_{\vec{Y}^*}} p(\vec{D}^* | \mathrm{do}(\mathrm{spa}(\vec{D}^*)))$$

Consider the following example, which illustrates how each term in this factorization is identified by one of three cases: either directly by the ID algorithm, or the gID algorithm, or a new case, formalized via Algorithm 2, which obtains identification via the most general version possible of the perfect instrument trick described in the previous section. Failure cases return either the hedge [Shpitser and Pearl, 2006], or the thicket [Lee et al., 2019].

---

**Algorithm 1:** SS-ID (systematic selection ID)

**Data:** $\mathcal{G}^{[]}, \vec{a}, \vec{Y}, p(\vec{V})$
**Result:** $p(\vec{Y}(\vec{a}, S = \emptyset))$ or FAIL

1   $\vec{Y}^* \leftarrow \mathrm{an}_{\mathcal{G}^{[]}(\vec{a}, \emptyset)}(\vec{Y})$ ;
2   **for** $\vec{D}^* \in \mathcal{D}(\mathcal{G}^{[]}_{\vec{Y}^*})$ **do**
3     **if** *no $s$ exists that is laidback for $\vec{D}^*$* **then**
4       **return** FAIL(positivity)
5     **if** $\mathrm{cl}(\vec{D}^*) = \vec{D}^*$ **then**
6       $q(\vec{D}^* | \mathrm{spa}(\vec{D}^*)) \leftarrow \phi_{\vec{V} \backslash \vec{D}^*}(p, \mathcal{G}^{[]})|_{S=s}$,
7       $s$ laidback for $\vec{D}^*$, consistent with $a_{\mathrm{spa}(\vec{D}^*)}$
8     **else**
9       $\tilde{\mathcal{G}} \leftarrow \phi_{\vec{V} \backslash \mathrm{cl}(\vec{D}^*)}(\mathcal{G}^{[]}); \tilde{q} \leftarrow \phi_{\vec{V} \backslash \mathrm{cl}(\vec{D}^*)}(p; \mathcal{G}^{[]})$;
10       **if** $S \notin \mathrm{cl}(\vec{D}^*)$ **then**
11         **if** *there is $s$ laidback for $\vec{D}^*$, consistent with $\vec{a}_{\mathrm{spa}(\vec{D}^*)}$, and $\vec{D}^*$ reachable in $\tilde{\mathcal{G}}^{[s]}$* **then**
12           $q(\vec{D}^* | \mathrm{spa}(\vec{D}^*)) \leftarrow \phi_{\mathrm{cl}(\vec{D}^*) \backslash \vec{D}^*}(\tilde{q}; \tilde{\mathcal{G}}^{[s]})$
13         **else**
14           **return** FAIL(thicket)
15       **else**
16         $q(\vec{D}^* | \mathrm{spa}(\vec{D}^*)) \leftarrow$ *Algorithm 2*$(\tilde{\mathcal{G}}, \vec{a}, \tilde{q}, \vec{D}^*, \mathrm{cl}(\vec{D}^*))$
17   **return** $\sum_{\vec{Y}^* \backslash \vec{Y}} \prod_{\vec{D}^*} q(\vec{D}^* | \mathrm{spa}(\vec{D}^*))|_{\vec{a}_{\vec{A} \cap \mathrm{spa}_{\mathcal{G}}(\vec{D}^*)}, s_{\vec{D}^*}}$,
18   with $s_{\vec{D}^*}$ laidback for $\vec{D}^*$, consistent with $\vec{a}_{\vec{A} \cap \mathrm{spa}_{\mathcal{G}}(\vec{D}^*)}$.

---

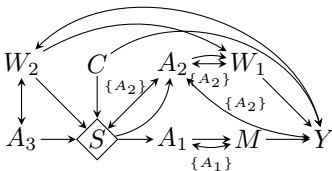

Figure 2: An LS-ADMMG illustrating Algorithm 1.

**Example 1** (Identifying $p(Y(\vec{a}, S = \emptyset))$ in Fig. 2).
*The identifying functional is* $p(Y(\vec{a}, S = \emptyset)) = \sum_{\vec{Y}^* \backslash Y} \prod_{D_i^* \in \mathcal{D}(\mathcal{G}^{[]}(a, \emptyset)_{\vec{Y}^*})} q_{\vec{D}_i^*}(\vec{D}_i^* | \mathrm{spa}(\vec{D}_i^*))$, *which is equal to*

$$\sum_{M, W_1, W_2, C} p(C) p(M | a_1, s_{a_1}) p(W_1 | W_2, a_2, s_{a_2})$$
$$\sum_{A_3} p(Y | M, W_2, W_1, C, s_{a_1, a_2}, A_3) p(W_2, A_3),$$

*where $s_{\vec{a}}$ is a shorthand for any value of $S$ which is serious for $\vec{A}$ at values $\vec{a}$. See the Appendix for a detailed derivation.*

Our results show that our proposed algorithm is sound, and implies a non-identified query in all but one failure cases. We illustrate a number of failure cases of the algorithm in the Appendix. We conjecture this algorithm is also complete.

**Theorem 4** (Soundness). *Algorithm 1 is sound.*

**Theorem 5** (Non-identification). *If Algorithm 1 fails at Algorithm 1, line 4, Algorithm 1, line 14, or Algorithm 2, line 3 then the causal effect is not identified.*

# 7 CONCLUSIONS

In this paper, we have considered the problem of identification of causal effects in settings with multiple datasets, corresponding to the observational or interventional contexts derived from a causal model where units are selected into different contexts *systematically*. Unlike prior approaches, we represent systematic selection by means of an indicator random variable that is potentially related to other variables in the model in complicated ways. We show that in the resulting *Context Selected Structural Causal Model (CS-SCM)* systematic selection may be arranged into a hierarchy resembling the hierarchy of systematic censoring in missing data, with possible models including selected completely at random (SCAR), selected at random (SAR), and selected not at random (SNAR). We show that in SCAR and SAR models, identification of interventional distributions may be obtained by a generalization of the g-formula.

**Algorithm 2:** Identification for a confounded selector

**Data:** $\mathcal{G}^{[]}(\vec{C}, \mathrm{spa}(\vec{C})), a, q_{\vec{C}}(\vec{C} \mid \mathrm{spa}(\vec{C})), \vec{D}^*, \vec{C};$
where $\vec{C} \equiv \mathrm{cl}(\vec{D}^*);$

**Result:** $q_{\vec{D}^*}(\vec{D}^* \mid \mathrm{spa}(\vec{D}^*))$ or FAIL

1   $\mathrm{ch}^*(S) \leftarrow \mathrm{ch}(S) \cap (\mathrm{cl}(\vec{D}^*) \setminus \vec{D}^*);$

2   **if** $\mathrm{ch}^*(S) = \emptyset$ **then**

3      **return** FAIL(hedge$\langle \vec{D}^*, \mathrm{cl}(\vec{D}^*)\rangle$);

4   **else**

5      **for** $\bar{s} \in \mathfrak{X}_S^+$ which are laidback for $\vec{D}^*$ but serious for $\vec{Z} \subseteq \mathrm{ch}^*(S)$ at $\vec{z}$ consistent for $a_{\mathrm{spa}(\vec{D}^*)}$ **do**

6        Let $\vec{D}' \in \mathcal{D}(\mathcal{G}^{[]}(\bar{s}))$, s.t. $\vec{D}^* \subseteq \vec{D}'$ ;

7        **if** $\{D(\bar{s}) : D \in \vec{D}' \cap \mathrm{de}(S)\} \perp\!\!\!\perp S \mid \{D(\bar{s}) : D \in \vec{D}' \cap \mathrm{nd}(S)\} \cup \mathrm{spa}(\vec{D}')$ in $\mathcal{G}^{[]}(\bar{s})$ and $D^*$ is reachable in $\mathcal{G}^{[s]}(\bar{s})_{\vec{D}'}$ **then**

8          $q_{\vec{D}'}^{s,\vec{z}}(\vec{D}' \mid \mathrm{spa}(\vec{D}')) \leftarrow$
$$\left[ \prod_{D \in \mathrm{de}(S) \cap \vec{D}'} q_{\mathrm{cl}(D^*)}(D \mid \bar{s}, \mathrm{pre}_\prec(D)) \right]\Big|_{\vec{Z}=\vec{z}}$$
$$\times$$
$$\left[ \prod_{D \in \mathrm{nd}(S) \cap \vec{D}'} q_{\mathrm{cl}(D^*)}(D \mid \mathrm{pre}_\prec(D)) \right]\Big|_{\vec{Z}=\vec{z}},$$
where $\mathrm{pre}_\prec(D)$ are topological predecessors of $D$ in $D' \cup \mathrm{spa}(D')$.
         **return** $\phi_{\vec{D}' \setminus D}(q_{\vec{D}'}^{s,\vec{z}}; \mathcal{G}^{[s]}(\bar{s})_{\vec{D}'});$

9      **return** FAIL;

In SNAR settings, where systematic selection and unobserved confounding are present, we provide a general identification algorithm which generalizes the gID algorithm [Lee et al., 2022, Kivva et al., 2022], but which applies in causal models with arbitrarily complex types of systematic selection, and is able to achieve novel identification results using context-specific restrictions found in CS-SCMs.

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

# A General Identification Algorithm For Data Fusion Problems Under Systematic Selection
## (Supplementary Material)

**Jaron J. R. Lee**[1]  **AmirEmad Ghassami**[2]  **Ilya Shpitser**[1]

[1]Department of Computer Science, Johns Hopkins University, Baltimore, Maryland, USA
[2]Department of Mathematics and Statistics, Boston University, Boston, Massachusetts, USA,

## A PROOFS

**Theorem 1.** *Given a CS-SCM associated with $\mathcal{G}^{[]}(\vec{V})$, and any $\vec{A} \subseteq \vec{V}$ such that $S \in \vec{A}$ (including $\vec{A} = \{S\}$), any d-separation statement in $\mathcal{G}^{[s]}(\vec{a})$, for s consistent with $\vec{a}$, implies a conditional independence statement in $p(\vec{V}(\vec{a}))$.*

*Proof.* This result is established by proof of soundness of the CSI-separation criterion in Boutilier et al. [1996], definition of CS-ICMs (specifically, definition of seriousness of $S$ with respect to any element of $\mathrm{ch}_{\vec{\mathcal{G}}}(S)$), and a direct extension of results in [Richardson and Robins, 2013b]. □

**Theorem 2** (Context Selected g-formula). *Fix a fully observed CS-SCM corresponding to an LS-DAG $\mathcal{G}^{[]}$ with a vertex set $\vec{V}$, and disjoint subsets $\vec{A}, \vec{Y}$ of $\vec{V}$. Let $\vec{Y}^* = \mathrm{an}_{\mathcal{G}^{[]}(\vec{a}, \emptyset)}(\vec{Y})$ Then $p(\vec{Y}(\vec{a}, S = \emptyset))$ is identified if and only if for every element $V \in \vec{Y}^*$ there exists a value $s_V \in \mathfrak{X}_S^+$ laidback for $V$ (i.e. $s_V^e = 0$). If so, we have:*

$$p(\vec{Y}(\vec{a}, S = \emptyset)) = \sum_{\vec{Y}^* \setminus \vec{Y}} \prod_{V \in \vec{Y}^*} p(V | \mathrm{pa}_{\mathcal{G}}(V))|_{\vec{a}_{\vec{A} \cap \mathrm{pa}_{\mathcal{G}}(V)}, S_V^e = 0}. \tag{2}$$

*Proof.* Soundness follows by application of the g-formula to the CS-SCM [Robins, 1986], and the definition of our query.

Completeness holds by the following argument. Assume that $S$ is never laidback for some element of $\vec{Y}^*$, which we call $Z$. Then, it is possible to produce two models which have different distributions on $Z$ when $S = \emptyset$. It is then possible to construct two models which agree on the observed data distribution (which only includes elements in $\mathfrak{X}_S^+$), but disagrees on $p(\vec{Y}(a, S = \emptyset))$. In particular, fix $Y \in \vec{Y}$. Since $Z \in \vec{Y}^*$, there must be a directed path from $Z$ to $Y$. Consider two elements of the causal model where the only edges are on the directed path from $Z$ to $Y$, and all vertices are otherwise mutually independent. Then it is straightforward to construct two elements of the causal model where the mapping from $p(Z)$ to $p(Y)$ given by $\sum_Z p(Y \mid Z)$ is one to one. Since the rest of the vertices of the model as mutually independent, we have that in the two elements we are considering, $p(\vec{Y}(a, S = \emptyset)) = \prod_{\tilde{Y} \in \vec{Y}} p(\tilde{Y})$. This immediately yields non-identification since we can construct two elements that agree on $p(\tilde{Y})$ for every $\tilde{Y} \in \vec{Y} \setminus \{Y\}$, and indeed on the observed data distribution, but disagree on $p(Y)$. □

**Theorem 6** (Hedge for $S = \emptyset$ interventions). *Let $\mathcal{G}$ be a graph with vertex set $\vec{V}$, with $S \in \vec{V}$ representing an CS-SCM. Let $\vec{F}', \vec{F}$ be bidirected-connected sets in $\mathcal{G}$ where $\vec{F} \subset \vec{F}'$, and $S \in \vec{F}' \setminus \vec{R}$, and $\vec{R}$ is the root set of both $\vec{F}$ and $\vec{F}'$. Then, $p(\vec{F} \mid \mathrm{do}(S = \emptyset))$ is not identified.*

*Proof.* We first begin by defining an edge subgraph $\mathcal{G}'$ of $\mathcal{G}$, in which we retain all vertices, all bidirected edges, all directed edges in $\mathrm{an}_{\mathcal{G}}(S)$, and all directed edges from $S$ to its children. Define $\vec{Z} = \mathrm{cl}_{\mathcal{G}'}(\vec{D}^*) \setminus \mathrm{an}_{\mathcal{G}'}(S)$.

We then consider the districts $\vec{D}' \in \mathcal{D}(\mathcal{G}'_{\vec{Z}})$. These districts are all bidirected connected components that have bidirected edges to $\mathrm{an}_{\mathcal{G}'}(S)$, and may or may not have children of $S$.

Choose a particular child of $S$, call it $L$. We consider $\vec{D}'_L \in \mathcal{D}(\mathcal{G}'_{\vec{Z}})$ which contains $L$.

Then, $L$ may be connected to $S$ in one of two ways by a bidirected path (which exists solely in $\vec{D}'_L \cup \mathrm{an}_{\mathcal{G}'}(S)$) - either this bidirected path enters $\mathrm{an}_{\mathcal{G}'}(S)$ via $S$, or via $\mathrm{an}_{\mathcal{G}'}(S) \setminus \{S\}$.

1. **Bidirected path enters $\mathrm{an}_{\mathcal{G}'}(S)$ via $S$**: In the first case, we can isolate the bidirected path from $L$ to $S$. Denote the variables on this path as $W_1, \ldots, W_k$, where some subset of these variables may also be children of $S$. Then it is possible to provide a modified hedge construction in which $p(L, \vec{W} \mid \mathrm{do}(S = \emptyset))$ is not identified.

   We first describe a process to augment the graph $\mathcal{G}'$, which we denote as $\mathcal{G}''$. $\mathcal{G}''$ inherits all vertices of $\mathcal{G}'$, but only edges present in the subgraph over $\mathcal{G}'_{\vec{W} \cup \{L,S\}}$. We then split $S$ into an separate nodes $S^e_{C_i}$ for each child $C_i$, and these $S^e_C$ form a bidirected chain $S^e_Y \leftrightarrow S^e_{C_1} \leftrightarrow \ldots S^e_{C_k}$. The last $S^e_{C_k}$ inherits the bidirected edge into $S$. For each child $C$, we replace the $S \to C$ edge with $S^e_C \to C$. For each child $C$, we create $S^v_C$ that has a single unobserved variable $U_{S^v_C}$ as parent, and has directed edge $S^v_C \to C$. Finally, we replace each bidirected edge $V \leftrightarrow W$ with $V \leftarrow U_{V,W} \to W$, where $U_{V,W}$ denotes an unobserved variable. $\mathcal{G}''$ is a subgraph of $\mathcal{G}'$ if you marginalize $\vec{U}$ and perform a cartesian product operation on all $\vec{S}^e = \{S^e_C\}_{C \in \mathrm{ch}_{\mathcal{G}'}(S)}, \vec{S}^v = \{S^v_C\}_{C \in \mathrm{ch}_{\mathcal{G}'}(S)}$.

   Given $\mathcal{G}''$, we now describe a procedure to construct two models respecting Definition 1 which agree on the observed distribution, but disagree on the causal effect $p(\vec{F} \mid \mathrm{do}(S = \emptyset))$. Let the cardinality of all variables to 2. In model 1, the value of each variable is equal to the bit parity of the parents. If $S^e_C$ is in the parents, then if $S^e_C = 0$ it takes on the bit parity of the other parents (noting that this is equivalent to the bit parity of all parents, since $0 \oplus X = X$ for any bit $X$), and if $S^e_C = 1$ then the variable takes on the value of $S^v_C$. The same is true in model 2, except $W_1$ does not pay attention to the bit connecting it and the last $S^e_C$.

   In the observational setting, in both models, the structural equations are the same for $\vec{S}^e, \vec{S}^v$. In model 1, when $\vec{S}^e = \vec{0}$, $L \cup \vec{W}$ effectively counts the bit parity of each $U$ in $\mathcal{G}''$ twice, since $S^e_L$ is zero only if the $U'$ connecting it to $W_1$ is zero. This forms a distribution where values of $L \cup \vec{W}$ which have even bit parity have equal probability, and there is no probability mass elsewhere. If any $S^e_C = 1$, then there exists at least one such $U$ which has only one path to $L \cup \vec{W}$ and so the distribution is uniform. In model 2, when $\vec{S}^e = 0$, $L \cup \vec{W}$ effectively counts the bit parity of each $U$ in $\mathcal{G}''_{L \cup \vec{W}}$ twice, ignoring the $U$ that connects $\vec{W}$ to $S$, resulting in a distribution where values of $L \cup \vec{W}$ which have even bit parity having equal probability and no mass elsewhere. As with model 1, when any $S^e_C = 1$ then the distribution becomes uniform. Thus the observed data distributions agree.

   Under an intervention $S = \emptyset$, this has the effect of ensuring that in model 2, the bit parity of $L \cup \vec{W}$ is always even, whereas in model 1 it is uniform, because there is the contribution of the $U'$. This establishes the non-identification of $p(\vec{D}'_L \mid \mathrm{do}(S = \emptyset))$.

2. **Bidirected path enters $\mathrm{an}_{\mathcal{G}'}(S)$ via $\mathrm{an}_{\mathcal{G}'}(S) \setminus \{S\}$**: Since $S$ must have at least one child, we select any child at random; call this child $L$. We consider the districts of $\mathcal{G}'_{\vec{Z}}$, and define $\vec{D}'_L$ to be the district containing $L$. Define $\vec{R}$ to be the root set of $\vec{F}$.

   We first begin by providing an augmented graph $\mathcal{G}''$ constructed from $\mathcal{G}'$. This graph retains all vertices of $\mathcal{G}'$, all children of $S$, all directed edges in the ancestors of $S$. We then split $S$ according to the following procedure: $S$ is replaced with vertices $S^e_C, S^v_C$ for each $C \in \mathrm{ch}_{\mathcal{G}'}(S)$. $S^e_L$ inherits all incoming arrowheads previously into $S$. We create directed edges $S^e_C \to C, S^v_C \to C$ for each child $C$. For all $S^v_C$, we add an unobserved variable $U_{S^v_C} \to S^v_C$, and for all $S^e_C$ which is not $S^e_L$, we add $U_{S^e_C} \to S^e_C$. We note that $\mathcal{G}''$ is an edge subgraph of $\mathcal{G}'$ if the $\vec{U}$ are latent projected, and the $\vec{S}^e, \vec{S}^v$ are grouped into a single vertex via the cartesian product operator.

   Next, we will now define a CS-SCM, by modifying the construction of Shpitser and Pearl [2006] in such a way to respect the context-specific restrictions.

   In model 1, if $V \notin \mathrm{ch}_{\mathcal{G}''}(S)$ then $V \equiv \oplus \mathrm{pa}_{\mathcal{G}''}(V)$. Otherwise, $V \equiv \begin{cases} \oplus \mathrm{pa}_{\mathcal{G}''}(V) & S^e_V = 0 \\ S^v_V & S^e_V = 1 \end{cases}$.

   In model 2, the same is true, except for variables in $\vec{D}'_L$. For those variables, if $V \notin \mathrm{ch}_{\mathcal{G}''}(S)$ then they only pay attention to parents in $\vec{D}'_L$, and otherwise $V \equiv \begin{cases} \oplus \mathrm{pa}_{\mathcal{G}''_{\vec{D}'_L}}(V) & S^e_V = 0 \\ S^v_V & S^e_V = 1 \end{cases}$.

   In the observational distribution, both models induce the same distribution. First, we point out that for variables outside of $\vec{D}'_L$, the structural equations (and therefore the parts of the distribution associated with those variables) are the same. Next, we consider variables in $\vec{D}'_L$. First we point out that the distributions over $\vec{S}^e$ are the same in both models,

and are uniform random distributions. When $\vec{S}^e = 0$, in model 1 variables in $\vec{D}'_L$ count the bit parity of their parents twice, whereas in model 2 variables in $\vec{D}'_L$ count the bit parity of their parents in $\vec{D}'_L$ twice. Either way this induces a conditional distribution (of $p(\vec{D}'_L \mid \vec{S}^e, \vec{S}^v)$) with equal probability over even bit parities, and zero probability otherwise. When there exists a child of $S$ such that $\vec{S}^e_C = 1$, then in both models the conditional distribution $p(\vec{D}'_L \mid \vec{S}^e, \vec{S}^v)$ is uniform because there exists at least one $U$ which has only one path down to $\vec{D}'_L$.

In the interventional distribution where the intervention $\mathrm{do}(S = \emptyset)$ is applied, the distributions in the two models differ. In model 1, the distribution over $\vec{D}'_L$ is uniform, because the $U$ variables which connect $\mathrm{an}_{\mathcal{G}''}(S)$ to $\vec{D}'_L$ now only have one path to $\vec{D}'_L$. In model 2, the distribution has equal mass assigned to even bit parities for $\vec{D}'_L$, and no mass to other values.

This establishes the non-identification of $p(\vec{D}'_L \mid \mathrm{do}(S = \emptyset))$.

If $\vec{R} \subseteq \vec{D}'_L$, then we have a witness for the non-identifiability of $p(\vec{R} \mid \mathrm{do}(S = \emptyset))$, and since $\vec{R} \subseteq \vec{F}$ this proves the claim.

Otherwise, since the intervention is $S = \emptyset$, all edges in the graph $\bar{\mathcal{G}}^{[]}$ may be used in our construction of counterexamples. In particular, we can employ the downward extension of Theorem 4 found in Shpitser and Pearl [2006] to both elements of the causal model we are constructing as counterexamples. This gives

$$p(\vec{R} \mid \mathrm{do}(S = \emptyset)) = \sum_{\vec{D}'_C} p(\vec{R} \mid \vec{D}'_C, S = \emptyset) p(\vec{D}'_C \mid \mathrm{do}(S = \emptyset))$$

where to suffices to choose $p(\vec{F} \mid \vec{R}, S = \emptyset)$ that will yield a one to one mapping from $p(\vec{D}'_C \mid \mathrm{do}(S = \emptyset))$ to $p(\vec{R} \mid \mathrm{do}(S = \emptyset))$ in the above equation.

$\square$

**Theorem 3.** *Given a hidden variable CS-SCM represented by a LS-ADMMG $\mathcal{G}^{[]}(\vec{V})$, the ID algorithm with causal query $p(\vec{V}|do(S{=}\emptyset))$, data distribution $p(\vec{V})$, and ADMG $\mathcal{G}^{[\emptyset]}$ is sound and complete.*

*Proof.* The ID algorithm is sound for queries from models in the CS-SCM. This is because the CS-SCM is a submodel (contains more restrictions) than the models considered in Shpitser and Pearl [2006], and the ID algorithm was established to be sound in that same paper.

To see that the ID algorithm is complete for this query, we need to establish that whenever the ID algorithm fails, we can construct two models which have the same observed data distribution but different counterfactual distributions for $p(\vec{V} \mid \mathrm{do}(S = \emptyset))$

As established in Shpitser and Pearl [2006], Richardson et al. [2023], the ID algorithm fails when there is a district $\vec{D}^* \in \mathcal{D}(\mathcal{G}_{\vec{V} \setminus \{S\}})$ whose closure $\mathrm{cl}_{\mathcal{G}}(\vec{D}^*)$ is such that $\vec{D}^* \subset \mathrm{cl}_{\mathcal{G}}(\vec{D}^*)$. Furthermore, we can establish that $S \in \mathrm{cl}_{\mathcal{G}}(\vec{D}^*) \setminus \vec{D}^*$, and that $\vec{D}^*$ must not be a district of $\mathcal{G}$, since otherwise $\vec{D}^*$ is reachable in $\mathcal{G}$.

When such a $\vec{D}^*$ is encountered in the process of running the ID algorithm, we will return a construction from Theorem 6. The original hedge construction of Shpitser and Pearl [2006] is not suitable because it does not incorporate the special behavior introduced via the $S$ context variable. This bears witness to the non-identifiability of $p(\vec{D}^* \mid \mathrm{do}(S = \emptyset))$.

Because $\vec{D}^* \subseteq \vec{V}$, it immediately follows that $p(\vec{V} \mid \mathrm{do}(S = \emptyset))$ is not identified.

$\square$

**Theorem 4** (Soundness). *Algorithm 1 is sound.*

*Proof.* The algorithm aims to identify $p(\vec{Y} \mid \mathrm{do}(\vec{a}, S = \emptyset))$ in the causal model $\mathcal{G}$, with additional restrictions pertaining to the semantics of $S$, and its relationship to its children in $\mathcal{G}$, from the observed data distribution $p(\vec{V})$. These restrictions do not affect district factorizations of the observed and interventional distributions which hold due to Shpitser and Pearl [2006], Richardson et al. [2023].

Then, for value assignment $v \in \mathfrak{X}_{\vec{V}}$,

$$p(\vec{Y} = v_{\vec{Y}} \mid \mathrm{do}(\vec{a}, S = \emptyset)) = \sum_{\vec{Y}^* \setminus \vec{Y}} \prod_{D \in \mathcal{D}_{\mathcal{G}_{\vec{Y}^*}}} p(v_D \mid \mathrm{do}(v_{\mathrm{spa}_{\mathcal{G}}(D)))),$$

where values $v_{\mathrm{spa}_{\mathcal{G}}(D)})$ in each term are consistent with $\vec{a}$ and $S = \emptyset$.

Each term is identified by one of three cases.

The first case is triggered at Algorithm 1, line 7. In this case, we are justified in the choice of using any laid-back value $s$ for $\vec{D}^*$, because either $S$ is independent of $\vec{D}^*$ given its Markov blanket, or because the structural equations are all the same under those values due to mechanism invariance implied by the definition of the CS-SCM (see the restriction on the structural equation when $S_V^e = 0$ in Definition 1). For that value of $s$, and kernels evaluated to that value, soundness follows by the standard soundness argument of the ID algorithm [Shpitser and Pearl, 2006, Richardson et al., 2023], which holds in any SCM with independent errors, and thus also in an CS-SCM.

The second case is triggered at Algorithm 1, line 11, where $\vec{D}^* \subset \mathrm{cl}_{\mathcal{G}}(\vec{D}^*)$ and $S \notin \mathrm{cl}_{\mathcal{G}}(\vec{D}^*)$. This follows from the soundness of the gID algorithm [Lee et al., 2019]. Specifically, this case shows that the distribution $p(\mathrm{cl}_{\mathcal{G}}(\vec{D}^*) \mid \mathrm{do}(\mathrm{spa}_{\mathcal{G}}(\mathrm{cl}_{\mathcal{G}}(\vec{D}^*))))$ is identified, and represents the observed data distribution corresponding to a causal model represented by graph $\mathcal{G}_{\mathrm{cl}_{\mathcal{G}}(\vec{D}^*)}$. Since $S \notin \mathrm{cl}_{\mathcal{G}}(\vec{D}^*)$, the available datasets in this model may be reformulated as observational and interventional distributions on $\mathrm{cl}_{\mathcal{G}}(\vec{D}^*)$ indexed by values of $S$. These are precisely the inputs of the gID algorithm, and soundness follows by the soundness of that algorithm.

The third case is triggered at Algorithm 1, line 16, where $\vec{D}^* \subset \mathrm{cl}_{\mathcal{G}}(\vec{D}^*)$ and $S \in \mathrm{cl}_{\mathcal{G}}(\vec{D}^*)$.

Pick an appropriate value $\bar{s}$. We note that $p(\vec{D}'(\bar{s}, \mathrm{spa}(\vec{D}')))$ is identified because of the following derivation:

$$
\begin{aligned}
p(\vec{D}'(\bar{s}, \mathrm{spa}(\vec{D}'), \vec{V} \setminus \mathrm{cl}_{\mathcal{G}}(D^*))) &= \prod_{D \in \vec{D}'} q_{\mathrm{cl}_{\mathcal{G}}(D^*)}(D(\bar{s}) \mid \{W(\bar{s}) : W \in \mathrm{pre}_{\prec}(D)\}) \\
&= \left( \prod_{D \in \vec{D}' \cap \mathrm{de}_{\mathcal{G}}(S)} q_{\mathrm{cl}_{\mathcal{G}}(D^*)}(D(\bar{s}) \mid \{W(\bar{s}) : W \in \mathrm{pre}_{\prec}(D)\}) \right) \times \\
&\quad \times \left( \prod_{D \in \vec{D}' \cap \mathrm{nd}_{\mathcal{G}}(S)} q_{\mathrm{cl}_{\mathcal{G}}(D^*)}(D(\bar{s}) \mid \{W(\bar{s}) : W \in \mathrm{pre}_{\prec}(D)\}) \right) \\
&= \left( \prod_{D \in \vec{D}' \cap \mathrm{de}_{\mathcal{G}}(S)} q_{\mathrm{cl}_{\mathcal{G}}(D^*)}(D \mid S = \bar{s}, \{W : W \in \mathrm{pre}_{\prec}(D)\}) \right) \times \\
&\quad \times \left( \prod_{D \in \vec{D}' \cap \mathrm{nd}_{\mathcal{G}}(S)} q_{\mathrm{cl}_{\mathcal{G}}(D^*)}(D \mid \{W : W \in \mathrm{pre}_{\prec}(D)\}) \right)
\end{aligned}
$$

where $\mathrm{pre}_{\prec}(D)$ is the subset of $\vec{D}' \cup \mathrm{spa}_{\mathcal{G}}(\vec{D}')$ earlier than $D$ under some ordering $\prec$ topological for $\mathcal{G}$. Here the first equality follows by the top level district factorization of any interventional distribution in an SCM, the second equality follows by arranging terms, and the third by independence, and rule 3 of the potential outcomes calculus [Malinsky et al., 2019].

Given this, soundness of the third case follows from soundness of the ID algorithm formulated via the fixing operator.

$\square$

**Theorem 5** (Non-identification). *If Algorithm 1 fails at Algorithm 1, line 4, Algorithm 1, line 14, or Algorithm 2, line 3 then the causal effect is not identified.*

*Proof.* To demonstrate non-identification at each of these failure points, we will provide a construction.

1. Algorithm 1, line 4: If there is no $s$ laidback for $\vec{D}^*$ then identification fails. This follows from the fact that identifying the joint distribution $p(\vec{D}^* \mid \mathrm{do}(\mathrm{spa}_{\mathcal{G}}(\vec{D}^*)))$ is impossible if only marginal distributions of this joint distribution are available and random variables in $\vec{D}^*$ do not exhibit additional factorization structure, since $\vec{D}^*$ is a district.

   Since joint distributions cannot be recovered from marginal distributions without further assumptions, we fail to identify $q_V(\vec{D}^* \mid \mathrm{spa}(\vec{D}^*))$. Recall that $S$ cannot be in $\vec{D}^*$ due to the definition of $\vec{Y}^*$.

   This case handles the degenerate identification failure case of the gID algorithm (e.g. Section 3.1 of Lee et al. [2019], or Section 3 of supplementary of Kivva et al. [2022], both provide details of explicit constructions showing this), where we may consider a model where $S$ has no parents and siblings to mimic the constructions.

2. Algorithm 1, line 14: A failure at this point involves the district $\vec{D}^*$ not containing $S$, but for each $s$ that is both consistent with $a_{\mathrm{spa}(\vec{D}^*)}$ and laidback for $\vec{D}^*$, $\vec{D}^*$ is not reachable in the corresponding $\mathcal{G}^{[s]}$. Then, we may adopt either the thicket construction of Lee et al. [2022] or corresponding alternative in Kivva et al. [2022], where $S$ is now viewed simply as an indexing operator for the various distributions that are inputs into gID. Then, the thicket construction immediately witnesses the non-identifiability of the desired causal effect $p(\vec{D}^* \mid \mathrm{do}(a, S = \emptyset))$.

3. Algorithm 2, line 3: Since $\mathrm{ch}^*(S)$ is empty, there are two possibilities – it is either the case that $S$ also has no children in $\vec{D}^*$, or $S$ has children in $\vec{D}^*$.

   In the first case the construction is simply given as a regular hedge per Shpitser and Pearl [2006], since $S$ has no children inside.

   In the second case, we will modify the argument of Shpitser and Pearl [2006] slightly in order to respect the constraints of Definition 1. For variables in $\mathrm{cl}(\vec{D}^*)$ which do not have $S$ as a parent, the structural equations are exactly as they appear in Shpitser and Pearl [2006] in both models. For other variables $V \notin \mathrm{ch}(S) \cap \vec{D}^*$, when $S$ is laidback for $V$ (meaning $S_V^e = 0$), the variable is equal to the bit parity of its parents as defined in Shpitser and Pearl [2006]. When $S$ is serious for $V$ (meaning that $S_V^e = 1$) the variable is equal to $S_V^v$ as required by Definition 1.

   This construction is a valid witness, and the proof is essentially the argument laid out in the second part of the proof for Theorem 6.

Given a non-identification structure for $p(\vec{D}^* \mid \mathrm{do}(a, S = \emptyset))$ in any of the above cases, we now consider an argument for showing that $p(\vec{Y} \mid \mathrm{do}(\vec{a}, S = \emptyset))$ is not identified.

We first note that $\vec{A} \cap \mathrm{spa}(\vec{D}^*)$ is guaranteed to be not empty, as this is the only way $\mathrm{cl}(\vec{D}^*) \subset \vec{D}^*$.

Let $\vec{R}^*$ be the set of variables with no children in graph $\mathcal{G}_{\mathrm{cl}(\vec{D}^*)}$. Let $\mathcal{G}'$ be the edge subgraph used in this construction, which consists of the edges in $\mathcal{G}_{\mathrm{cl}(\vec{D}^*)}$, and a subset of edges in $\mathcal{G}_{\vec{Y}^*}$ that form a forest from the root down to $\vec{Y}$. Let $\vec{Y}' \subseteq \vec{Y}$ be the subset of $\vec{Y}$ where the forest connects.

If $\mathrm{ch}(S)$ are not in this forest the original construction as detailed in Theorem 4 of Shpitser and Pearl [2006] holds.

Otherwise, without loss of generality, we may assume that there exists a value $s' \in \mathfrak{X}_S$ such that the forest connecting $\vec{R}$ to $\vec{Y}'$ is laid-back. Variables along the forest are equal to the bit parity of their parents if $S$ is laid-back for that variable, and equal to the suitable bit of $S$ if $S$ is serious for that variable. Then, we employ in that value a suitable one-to-one construction [Shpitser and Pearl, 2006, Lee et al., 2019] in both models. This gives

$$p(\vec{Y}' \mid \mathrm{do}(\vec{a}, S = \emptyset)) = \sum_{\vec{D}^*} p(\vec{Y}' \mid \vec{R}, \mathrm{do}(S = \emptyset)) p(\vec{R} \mid \mathrm{do}(a, S = \emptyset))$$

For all other variables in $\mathcal{G}$ that do not appear in the forest, we may assume uniform distributions that are identical in both models. Then, for values $s$ which are laid-back for the forest, $p(\vec{Y}' \mid \vec{R}, \mathrm{do}(S = \emptyset))$ will disagree between the two models, and for other values of $S$, they will agree (at least from the earliest serious variable onwards). However, this still suffices to prove that $p(\vec{Y}' \mid \mathrm{do}(\vec{a}, S = \emptyset))$ is not identified. Since this is a margin of $p(\vec{Y} \mid \mathrm{do}(a, S = \emptyset))$, this proves that the latter is also not identified.

Finally, we note that if we didn't have positive support on the value $s'$, then this would correspond to having less data available, and the causal target would still not be identified with less data.

$\square$

## B EXAMPLES

### B.1 FULL EXAMPLE

**Example 1** (continuing from p. 8). *Throughout this example, we will use the shorthand $s_{\vec{a}}$ to mean any value of $S$ where $\{s_A^e = 1, s_A^v = \vec{a}_A : A \in \vec{A}\}$.*

*We first note that $\vec{Y}^* = \{Y, M, W_1, W_2, C\}$, and $\mathcal{D}(\mathcal{G}_{\vec{Y}^*}^{[]}) = \{\vec{D}_1^* = \{Y, W_2\}, \vec{D}_2^* = \{M\}, \vec{D}_3^* = \{W_1\}, \vec{D}_4^* = \{C\}\}$.*

$\vec{D}_1^*$ *invokes Algorithm 1, line 16. Per Algorithm 2, line 6, the relevant district of $\mathcal{G}^{[]}(s)$ encapsulating $\vec{D}_1^*$ is $\vec{D}' = \{Y, A_3, W_2\}$, for $s = s_{a_1, a_2}$.*

$\vec{D}'$ *triggers Algorithm 2, line 7, since $Y(a_1, a_2) \perp\!\!\!\perp S \mid M(a_1), W_1(a_2), W_2, A_3$ and $\vec{D}_1^*$ is reachable in $\mathcal{G}^{[s]}(s)_{\vec{D}'}$. Then, $q_{\vec{D}'}^{\bar{s}, \vec{z}}(\vec{D}' | \operatorname{spa}(\vec{D}')) = p(Y | M, W_2, W_1, C, s_{a_1, a_2}, A_3) p(W_2, A_3)$, and $q_{\vec{D}_1^*}(\vec{D}_1^* | \operatorname{spa}(\vec{D}_1^*)) = \phi_{A_3}(q_{\vec{D}'}^{\bar{s}, \vec{z}}(\vec{D}' | \operatorname{spa}(\vec{D}')); \mathcal{G}^{[\bar{s}]}(\bar{s})_{\vec{D}'})$
$= \sum_{A_3} p(Y | M, W_2, W_1, C, s_{a_1, a_2}, A_3) p(W_2, A_3)$.*

$\vec{D}_2^*$ *reaches Algorithm 1, line 11, with $\tilde{q} = \phi_{\vec{V} \setminus \{M, A_1\}}(p(\vec{V}); \tilde{\mathcal{G}}^{[]}) = p(M, A_1 | S)$, yielding $q_{\vec{D}_2^*}(\vec{D}_2^* | \operatorname{spa}(\vec{D}_2^*)) = \phi_{A_1}(\tilde{q}; \tilde{\mathcal{G}}^{[s_{a_1}]})|_{a_1} = p(M | a_1, s_{a_1})$.*

$\vec{D}_3^*$ *reaches Algorithm 1, line 16 with $\bar{q} = q_{\{W_1, A_2, S\}}(W_1, A_2, S | W_2, A_3, C)$ and the corresponding LS-CADMMG. Let $s = s_{a_2}$. Per Algorithm 2, line 6, the district of $\mathcal{G}^{[]}(s)$ encapsulating $\vec{D}_3^*$ is $\vec{D}' = \{W_1\}$. $\vec{D}'$ triggers Algorithm 2, line 7, because $W_1(a_2) \perp\!\!\!\perp S | W_2, A_3, C$ in $\bar{q}$ and $\vec{D}_3^*$ is (trivially) reachable in $\mathcal{G}^{[s]}(s)_{\vec{D}'}$. Then, $q_{\vec{D}'}^{\bar{s}, \bar{z}}(\vec{D}' \mid \operatorname{spa}(\vec{D}')) = p(W_1 \mid W_2, a_2, s)$.*

$\vec{D}_4^*$ *reaches Algorithm 1, line 7, giving $q_{\vec{D}_4^*}(\vec{D}_4^* | \operatorname{spa}(\vec{D}_4^*)) = p(C)$. This is a case our algorithm has in common with the ID algorithm, as no special structure of the problem involving the selector $S$ needs to be involved.*

*The identifying functional is $p(Y(\vec{a}, S = \emptyset)) = \sum_{\vec{Y}^* \setminus Y} \prod_{D_i^* \in \mathcal{D}(\mathcal{G}_{\vec{Y}^*})} q_{\vec{D}_i^*}(\vec{D}_i^* | \operatorname{spa}(\vec{D}_i^*))$, which is equal to*

$$\sum_{M, W_1, W_2, C} p(C) p(M | a_1, s_{a_1}) p(W_1 | W_2, a_2, s_{a_2})$$
$$\sum_{A_3} p(Y | M, W_2, W_1, C, s_{a_1, a_2}, A_3) p(W_2, A_3)$$

### B.2 ILLUSTRATING THE USE OF MULTIGRAPHS FOR REPRESENTING LATENT PROJECTIONS OF LS-DAGS

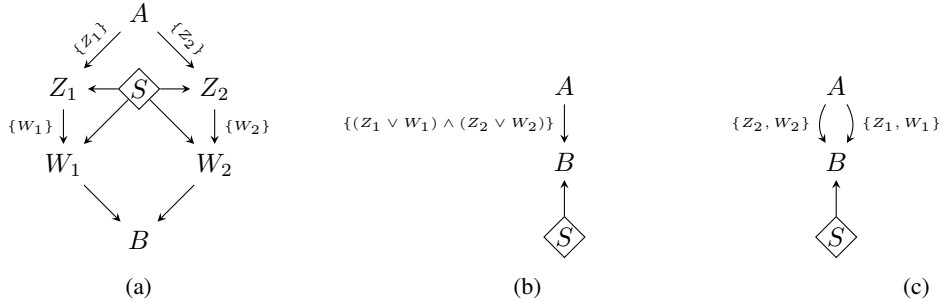

(a)  (b)  (c)

Figure 3: An LS-ADMMG illustrating the need for multigraph representations.

In this section we provide some further details on why multigraphs are required, especially under latent projections. Consider Fig. 3a. In a latent projection of $Z_1, Z_2, W_1, W_2$, only variables $A, B, S$ remain. The directed edge $A \to B$ can now disappear if we consider an intervention where at least one of $Z_1, W_1$ is intervened upon, and at least one of $Z_2, W_2$ is intervened upon. This could be represented by a boolean logic label (e.g. $\{(Z_1 \lor W_1) \land (Z_2 \lor W_2)\}$). However, computing the correct $\mathcal{G}^{[s]}$ graph then requires of a boolean expression on each label, which is difficult to interpret visually (see Fig. 3b).

Instead, we provide an alternative representation of the latent projection of Fig. 3a via a multigraph shown in Fig. 3c. Here, if given a value of $s$, the corresponding $\mathcal{G}^{[s]}$ graph can be recovered by checking for each edge whether the label intersects the value of $s$.

## B.3   FAILURE CASES

In this section we illustrate various failure cases that could occur throughout the application of Algorithm 1.

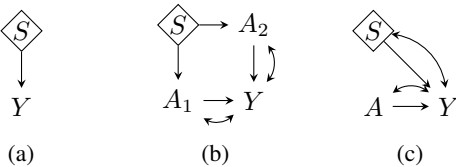

(a)                (b)                (c)

Figure 4: Examples of Failure Cases in Algorithm 1 and Algorithm 2

In all cases, we are interested in identifying a kernel for a particular district $\vec{D}^*$, which is in the graph $\mathcal{G}_{\vec{Y}^*}$, where we remind the reader that $\vec{Y}^* = \mathrm{an}_{\mathcal{G}_{\vec{V} \setminus (\vec{A} \cup \{S\})}}(\vec{Y})$.

1. Algorithm 1, line 4: Consider Fig. 4b, where we let $\mathfrak{X}_{S_0} = \{\{A_1\}, \{A_2\}\}$. Then, under each value of $S$, the district $\{Y\}$ is not reachable. A thicket can be constructed according to Lee et al. [2019], Kivva et al. [2022].

2. Algorithm 1, line 14: Consider Fig. 4a, where $\mathfrak{X}_{S_0} = \{\{Y\}\}$, and we are interested in identifying $p(Y \mid \mathrm{do}(S = \emptyset))$. Then in this case we never observe the true distribution of $Y$ where $S$ is laid-back for $Y$. Then it is easy to conceive of two models which have different distributions for $p(Y \mid S = (\emptyset, \emptyset))$, which is not part of observed data. The observed data in this case is the randomization probabilities on $Y$ that was specified by the experimenter.

3. Algorithm 2, line 3: Consider Fig. 4c, where $\vec{D}^* = \{Y, S\}$, and let $\mathfrak{X}_{S_0} = \{\emptyset, \{Y\}\}$. We see that Algorithm 1, line 16 gets triggered because there is an $s$ which is laid-back for $\vec{D}^*$ (namely $\emptyset$), $A$ is not fixable so $\mathrm{cl}(\vec{D}^*) = \{Y, A, S\} \neq \{Y, S\} = \vec{D}^*$, and that $S \in \mathrm{cl}(\vec{D}^*)$.

   Then, since $S$ is not a parent of $A$, the possibly modified hedge $\langle \{S, Y\}, \{S, Y, A\} \rangle$ is returned. See Theorem 5 for details.

## C   ANALYSIS

### C.1   COMPLEXITY OF ALGORITHM 1

As with all identification algorithms in graphical models, we may ask two distinct computational complexity questions.

The first question treats the algorithm as a decision procedure answering a YES/NO question about identification of a given query in a given model (and potentially giving additional useful information, such as the identifying functional if the answer is YES). The computational complexity of this version of our algorithm has an upper bound of a low order polynomial in the number of edges $|E|$ and vertices $|V|$ of the input graph. Specifically, here are the computational complexity calculations of a number of operations that appear in the algorithm: the computation of the set $\vec{Y}^*$ is $O(|E| + |V|)$, for which a depth-first traversal of the graph can calculate this set. The computation of districts in $\vec{Y}^*$ is similarly $O(|E| + |V|)$, via a depth first traversal. Positivity ("laidbackness") checks are linear in $\mathfrak{X}_S$, the size of the state space of S. Finding a fixable vertex is at most quadratic in $|V|$, and may be linear with clever use of hashing (since we must intersect descendants and districts). Iteration of the fixing operation happens at most $O(|V|)$ times. Calculations of reachability of the set entails iterative fixing. The overall algorithm is thus at most $O((|V| + |E| + |S|)^3)$, and more efficient implementations are likely possible.

The second computational complexity question is the time it takes to evaluate the identification query itself given, for example, categorical data. This version of the algorithm is likely intractable for the simple reason that the sum over $\vec{Y}^* \setminus \vec{Y}$ in front of the algorithm may be difficult to evaluate in general graphs, for much the same reason that variable elimination and belief propagation algorithms are intractable in dense graphs. While some prior work exists on making this version of the algorithm tractable using ideas from variable elimination algorithms [Shpitser et al., 2011], this work only applies to certain types of sparse graphs, and has not yet been generalized to the algorithm we present (although this is a very interesting area of future work).