# OpenReview forum: "A  General Identification Algorithm For Data Fusion Problems Under Systematic Selection"
_auai.org/UAI/2024/Conference — UAI 2024 oral_

### Official Review · Reviewer_sTkU · 2024-03-11

**Q2-1 Originality-Novelty:** 3
**Q2-2 Correctness-Technical Quality:** 3
**Q2-5 Clarity Of Writing:** 4

**Q10 Ethical Concerns:**

There are no ethical concerns.

**Q1 Summary And Contributions:**

**Short Summary:**

This paper considers the causal effect identification problem in the multi-dataset setting, where different datasets may correspond to the interventional or observational setting.
It presents a provably sound (but only conjecturally complete) algorithm for identifiability, proceeding from the g-formula used in the ID and gID algorithms.

**Long Summary**

 The multi-dataset setting considered in the paper is formalised by introducing a context selected SCM (CS-SCM) which augments the original SCM with selector variables (sometimes called the context variables), and different datasets correspond to different values of the selector variable.

The selection may be systematic, by which the authors mean that the selector variable may be influenced by other observed or unobserved variables.
In other words, the authors consider the setting where the selector variable may either be exogenous ("selected completely at random"),  or endogenous and influenced only by observed variables ("selected at random") or endogenous and influenced by observed and unobserved variables ("selected not a random"). In making such a distinction, the papers draws a parallel between the hierarchy of systematic selection it presents and the missing data hierarchy of Rubin'76.

Finally, the paper presents a general identification algorithm for answering interventional queries, given the multigraph corresponding to the CS-SCM, and the probability distribution corresponding to the mutiple datasets. It attributes the failure of identifiability to (i) a failure of positivity, or (ii) a thicket construction or (iii) a hedge construction.

**Q2-3 Extent To Which Claims Are Supported By Evidence:**

3: Good: the main claims are supported by convincing evidence (in the form of adequate experimental evaluation, proofs, (pseudo-)code, references, assumptions).

**Q2-4 Reproducibility:**

3: Good: key resources (e.g. proofs, code, data) are available and key details (e.g. proofs, experimental setup) are sufficiently well-described for competent researchers to confidently reproduce the main results.

**Q3 Main Strengths:**

1. This is a theoretically rich paper, which concisely explains the background material needed to understand the novel results on identifiability in the multi-dataset setting presented in Theorems 1, 2 and 3.

2. The papers presents a systematic selection ID (SSID) algorithm, which can have high impact in identifying causal effects in the systematic selection setting which was not protocolised thus far.

**Q4 Main Weakness:**

In some places, notations get overloaded which impedes the reading flow.
For instance, when defining fixable vertices for a CADMG, the vertex is chosen to be $V$, where another letter might be preferable, given the similarity to the set of vertices $\vec{V}$. The graph $\mathcal{G}$, is presented with and without an overbar (a distinction that isn't made sufficiently clear), and with and without $mathcal$.

**Q5 Detailed Comments To The Authors:**

**General Comments:**

1. Examples for the definitions can be introduced to ease comprehension, for instance, for the labelled selection DAG.
2. In the definition CS-SCM, if the selector variable $S$ is random variable, possibly endogenous due to the SAR and SNAR setting, where is its structural equation?



**Specific Comments:**

1. In Section 3, paragraph 8, where $\phi_V (q_{\vec{V}};\mathcal{G})$ is defined, it should probably say $V$ instead of $W$ after the conditioning symbol "$|$" in the kernel $q_{\vec{V}\setminus \{V\}} (\cdot)$.
2. In Section 4, paragraph 3: $\vec{j}_{D}$ is not defined.
3. Two types of citations appear for the same paper 'General Identifiability with Arbitrary Surrogate Experiments' by Lee et al.
4. The work on Joint Causal Inference (Mooij et al, JMLR 2020) merits a mention given its contribution to systematically defining the joint-SCM (over the system and context variables), and establishing assumptions required for joint causal discovery.
5. I would recommend to either use selector variable or context variable throughout the paper and specify in the beginning that these are used interchangeably in the literature.
6. In Appendix B.3, point 2, why is  $S$ a 2-tuple $ (\emptyset, \emptyset)$?
7. In Appendix B.3, point 3, first line, $\vec{D}^* = \{Y,S\}$ and not just $\{Y\}$.


**Typos/Grammar/Notational Issues:**

1. In Section 5, paragraph 5, there is a grammar issue within the quotes. One correct expression would be "the distributions of the outcomes $\vec{Y}$, when the context of the CS-SCM is set to the observational value and the variable $\vec{A}$ is set to $\vec{a}$. "
2. In Section 5.4, paragraph five, the last sentence should say "The third equality" instead of the fourth equality.
3. In appendix A, proof of Theorem 4, second-to-last line, it should say "do-calculus".

**Q9 Complying With Reviewing Instructions:**

Yes

---

> ### Author Rebuttal · Authors · 2024-04-05
>
> We thank the reviewer for their positive comments about this paper.
> We note that while we have not (yet!) proven completeness of our algorithm, we do provide strong partial results towards this goal in the sense that all failure cases of our proposed algorithm but one imply the resulting query is not identified.
> Regarding the question of notation, we are sympathetic that this is not an easy paper to read. We will make the suggested typographical and notational changes.
> Regarding the question about the selector, we intended for S to be grouped with variables in V \ ch(S) in the sense of having an arbitrary structural equation in terms of its parents.  However, since S is not present in the original graph or original SCM, it does not “inherit” the structural equation from the original model.  Instead, its structural equation is a part of the specification of the context selected SCM.  We will make this clearer – thank you for pointing this out.
> Finally we thank the reviewer for pointing out various typos, which will be fixed.

---

### Official Review · Reviewer_ATBw · 2024-03-19

**Q2-1 Originality-Novelty:** 3
**Q2-2 Correctness-Technical Quality:** 3
**Q2-5 Clarity Of Writing:** 3

**Q10 Ethical Concerns:**

No.

**Q1 Summary And Contributions:**

This paper proposes a new graphical identification algorithm for causal queries under selection. It builds on the recent results conditional acyclic directed mixed graphs (CADMGs) for the ID algorithm, extends these results to cover the gID algorithm and demonstrates the generality of the proposed algorithm, which is shown to be sound. The papers also makes an analogy between systematic selection and the missing data literature.

**Q2-3 Extent To Which Claims Are Supported By Evidence:**

4: Excellent: all claims are supported by very convincing evidence (in the form of comprehensive experimental evaluation, rigorous mathematical proofs, detailed (pseudo-)code, precise references, well-motivated and realistic assumptions) and the authors deliver what they promise.

**Q2-4 Reproducibility:**

4: Excellent: key resources (e.g. proofs, code, data) are available and key details (e.g. proof sketches, experimental setup) are comprehensively described for competent researchers to confidently and easily reproduce the main results.

**Q3 Main Strengths:**

The paper offers an elegant and succinct identification formula under selection, which echoes the work on CADMG. The authors clearly have a good grasp of the literature and provided enough review of the background for readers to catch up with the framework. The proposed algorithm is fairly general and covers many scenarios in selection. The claims are well supported by theory.

**Q4 Main Weakness:**

The work provides only a piece of pseudocode for the identification but no implementation nor its the time and space complexity. There is only one worked-out example following the steps of the proposed algorithm.

**Q5 Detailed Comments To The Authors:**

1. The motivation for the new identification algorithm is the need to define an appropriate superpopulation such that selection is well-defined in a probability model. It appears that the superpopulation is defined from intervention $\mathrm{do}(S=\emptyset)$ and not necessarily on a fully observational regime. Perhaps this can be made clearer?
1. The selection variable $S$ is a set of tuples $\langle S^e_V,S^v_V \rangle$. I found it difficult to follow the notation $S^v_V$. I feel like it should be defined more similar to a structural equation to replace the original one when $S^e_V$ is one. Leaving it as a random variable is not wrong, but can be confusing to others too. A reprise of the example used by Figures 1a-1b in the context of Figure 1c could also be very helpful.
---
1. The definition of district has a typo: $\mathrm{dis}_{\mathcal{G}}(X)=\lbrace{\color{red}X}\mid ...\rbrace\cup\lbrace X\rbrace$.
1. The symbols used for DAG with vertices $\vec{V}$ are not the same in Definition 1 ($\mathcal{G}(\vec{V})$) and Definition 2 ($\bar{\mathcal{G}}(\vec{V})$).
1. In the first paragraph of the second column on page 5, "$S$ is serious for $V$" should be "$s$ is serious for $V$".
1. There is a misspelt "everuy".

**Q9 Complying With Reviewing Instructions:**

Yes

---

> ### Author Rebuttal · Authors · 2024-04-05
>
> We thank the reviewer for the comments. We can provide some details regarding space and time complexity, which we did in response to Reviewer 2Y3A. We reproduce our response again here:
>
>
> Regarding the last point, we can provide a description of the time and space complexity of the algorithm. As with all identification algorithms in graphical models, we may ask two distinct computational complexity questions.  One would treat the algorithm as a decision procedure answering a YES/NO question about identification of a given query in a given model (and potentially giving additional useful information, such as the identifying functional if the answer is YES).  The computational complexity of this version of our algorithm has an upper bound of a low order polynomial in the number of edges |E| and vertices |V| of the input graph.  Specifically, here are the computational complexity calculations of a number of operations that appear in the algorithm:
> the computation of the set Y* is O(|E| + |V|), for which a depth-first traversal of the graph can calculate this set.
> the computation of districts in Y* is similarly O(|E| + |V|), via a depth first traversal.
> positivity (“laidbackness”) checks are linear in |S|, the size of the state space of S.
> finding a fixable vertex is at most quadratic in |V|, and may be linear with clever use of hashing (since we must intersect descendants and districts).
> Iteration of the fixing operation happens at most O(|V|) times.
> Calculations of reachability of the set entails iterative fixing.
> The overall algorithm is thus at most cubic in |V| + |E| + |S|, and more efficient implementations are likely possible.
> The second computational complexity question is the time it takes to evaluate the identification query itself given, for example, categorical data.  This version of the algorithm is likely intractable for the simple reason that the sum over Y* \ Y in front of the algorithm may be difficult to evaluate in general graphs, for much the same reason that variable elimination and belief propagation algorithms are intractable in dense graphs.  While some prior work exists on making this version of the algorithm tractable using ideas from variable elimination algorithms [1], this work only applies to certain types of sparse graphs, and has not yet been generalized to the algorithm we present (although this is a very interesting area of future work).
>
>
> We carefully chose the worked out example to exercise all parts of the algorithm, but can certainly add some simpler examples for the reader to follow.
>
> Regarding the first comment, during the development of our manuscript, we considered targets under the $\textrm{do}(S=\emptyset)$ regime versus the fully observational regime, both of which yield potentially interesting parameters It is tempting to consider targets in the fully observational regime. However, when one admits the selection variable as part of the causal model, the observational regime (where S is allowed to assume its natural value) yields parameters that effectively summarize/average over all domains under study, which can yield  strange and unintuitive results.
>
> Consider for a moment a model where $S \in \{\emptyset, <Y, 0>, <Y, 1> \}$. This represents an “observational” domain (e.g. a hospital ward which is functioning normally without intervention) and an “experimental” domain (e.g. a randomized controlled trial, where our outcome of interest $Y$ happened to be the target of intervention in that RCT, and we assume $Y$ is binary here). If we were interested in the target $p(Y)$ this would actually entail a weighted average of the observed probability distribution of $Y$ in the observational domain, and the experimenter determined randomization probabilities of $Y$ in the experiment. This is a well-defined target, but we judged that this is not very interesting to most analysts, as the target is now a function of randomization probabilities (which are determined by the experimenter and not intrinsic to causal mechanisms at play).
> If we had instead defined the target $p(Y \mid \textrm{do}(S=\emptyset)$, then this would correspond to the causal target only in the observational domain, meaning that we would compute a marginal distribution only using data from that domain. The experiment would not contain any information about the natural distribution of $Y$ under no intervention, and so we would discard it during identification. We judged that this target is closer in spirit to the original targets considered in the gID problem, and implicitly what analysts are after in the data fusion literature (the causal effect of variables A in the settings where all variables other than those in A are unaltered by the investigators). The need to specify the intervention $\textrm{do}(S=\emptyset)$ to represent this is a direct result of incorporating $S$ into the observed data distribution.

---

### Official Review · Reviewer_S8NW · 2024-03-20

**Q2-1 Originality-Novelty:** 2
**Q2-2 Correctness-Technical Quality:** 2
**Q2-5 Clarity Of Writing:** 2

**Q1 Summary And Contributions:**

This work provides an algorithm to deal with general identification with data fusion in the presence of selection bias. A framework similar to the hierarchy in the missing data problem has been adopted.

**Q2-3 Extent To Which Claims Are Supported By Evidence:**

2: Fair: the main claims are somewhat supported by evidence (but the experimental evaluation may be weak, or does not match entirely with the claims, important baselines may be missing, proofs contain important ideas but lack rigor, algorithmic details are only discussed superficially, references are imprecise, assumptions are not sufficiently motivated or explicated, etc.).

**Q2-4 Reproducibility:**

2: Fair: key resources (e.g. proofs, code, data) are unavailable but key details (e.g. proof sketches, experimental setup) are sufficiently well-described for an expert to confidently reproduce the main results.

**Q3 Main Strengths:**

1. The considered problem is practical and interesting.

2. The relation between the proposed method and others is introduced in a very clear way.

**Q4 Main Weakness:**

1. The theoretical novelty does not seem to be very significant. As suggested in the paper, the hierarchy of systematic selection is not novel since it was almost identical to that in the missing data problem. Thus, the main novel part of the setting lies in the introduction of a selector variable in the graphical structure. However, this is somehow similar to related works that introduce additional (random) variables as indicators of selection, intervention, distribution changes, missingness, or others. Many of these settings also allow the causal dependencies between the introduced variable and others. As a result, the proposed graphical causal model might not be novel enough.

2. Since an identification algorithm has been proposed, it would be helpful if some experimental results could be provided to validate its correctness. Although real-world experiments might not be necessary for a paper focusing more on theory, at least some basic simulations could be done to better verify the developed theory. Currently, the lack of that basic validation could be considered a weakness, since only showing several failure cases does not contribute a lot to a further understanding of the algorithm in general scenarios. Also, some simulation results could be helpful to see how likely the conjecture of completeness is true, which is important to evaluate the proposed algorithm.

3. Some key statements might not have rigorous support or the description might be misleading. Please refer to the detailed comments and questions.

**Q5 Detailed Comments To The Authors:**

1. Why Theorem 3 implies that $p(Y, A|do(S=\emptyset))$ is not identified?

2. In the introduction, it is emphasized that a sequential strategy to deal with selection and confounding cannot be complete. However, it seems that there is no rigorous proof of it. Please let me know if I missed anything.

3. Some statements do not seem to be rigorous and might lead to confusion. For example, in Section 5.4., it is stated that:

- ''However, identification **may** be obtained in an CS-SCM due to the following simple derivation ...''
- ''This derivation **may** be explained by noting that S acts as a perfect instrument ...''

4. The writing could be clearer. For instance, it might be better to make sections 2, 3, and 4 more succinct, since they are not original contributions of this work but take nearly half of the space.

5. What are the potential limitations of the proposed algorithm?

**Q9 Complying With Reviewing Instructions:**

Yes

---

> ### Author Rebuttal · Authors · 2024-04-05
>
> We thank the reviewer for their comments.
>
> We address the first critique that the reviewer presented in the main weaknesses. Respectfully, we disagree with this assessment.  One way to see the novelty is to compare the results in our work on graphical models of selection with prior work on graphical models of missing data [4,5].  Despite the fact that both selection problems and missing data problems feature missing completely at random, missing at random, and missing not at random hierarchy, the way in which systematic censoring and systematic selection are addressed in identification theory is quite different.  In addition, the work in causal inference most closely related to ours, specifically selection diagrams in the work of Pearl, Barenboim, Tian, and others, and graphs used to define decision theoretic causal models by Dawid, features selectors that are explicitly not random variables.  Finally, the proposed graphical model is only a part of our novel contribution in this paper, as we proceed to give a sound algorithm for inference that establishes identification under this kind of selection, and indicate that it subsumes earlier work on graphical data fusion. Our results show that dealing with selection and unobserved confounding is quite involved, which is demonstrated by a significantly increased methodological complexity of our algorithm compared to the prior state of the art, the gID algorithm.
>
>
> We now turn to the second critique. Prior influential work on identification theory, e.g. Pearl, Tian, Huang and Valtorta, Shpitser, Barenboim, Mohan, and others, generally did not provide statistical inference in the same paper, for two reasons.  First there is simply not enough room in a conference paper, once the theoretical underpinnings are established via proofs and mathematical argument. For instance, we are aware of multiple award-winning identification papers that have been published, that contain no experiments whatsoever, but do contain mathematical proofs supporting their claims.
> I. Shpitser and J. Pearl, “Identiﬁcation of Conditional Interventional Distributions,” UAI, 2006. - Best student paper award
> S. Lee, J. D. Correa, and E. Bareinboim, “General Identiﬁability with Arbitrary Surrogate Experiments,” Proceedings of the Conference on Uncertainty in Artificial Intelligence, p. 10, 2019. - Best paper award
> E. Bareinboim and J. Tian, “Recovering Causal Effects from Selection Bias,” Proceedings of the AAAI Conference on Artificial Intelligence, p. 7, 2015. - Best paper award
> Second, identification theory algorithms in causal inference and missing data by itself does not immediately lead to reasonable statistical procedures. Parametric maximum likelihood inference for outputs of the ID algorithm that capture the structure the causal model imposes on the observed data distribution required the development of appropriate likelihoods for discrete [1] and Gaussian [6] data.  Semi-parametric inference, which is an appropriate strategy if flexible likelihoods with infinitely many parameters are required, would entail development of estimators based on influence functions [2].  All these developments, while a definite direction for future work, are simply out of scope for our paper, which was already viewed as fairly ‘dense’ by peer review feedback.
>
> We now turn to the detailed comments.
>
> Regarding the first point - Theorem 3 allows us to show that the causal target $p(Y, A | \textrm{do}(S = \emptyset)) = p(\vec{V} | \textrm{do}(S=\emptyset))$ is not identified, by applying the ID algorithm [3].
>
> Regarding the second point, we demonstrated that completeness cannot hold because of the example shown in Fig 1f, for which the sequential strategy fails, but is in fact identifiable. Specifically the graph in Figure 1 f is an example of a so called “C-tree” or “convergent arborescence” [3].  In this graphical structure it is known that the effect of any variable or set of variables on the childless variable in the graph (in our case Y), is not identified by the ID algorithm (and thus not identified non-parametrically).  Thus, the effect of S on A,Y, obtained from p(Y,A | do(s)) is not identified, and the effect of A on Y is also not identified.  However, the derivation we present based on treating S as a perfect instrument of A is able to obtain identification by exploiting the special relationship of S and A implied by the context-selected structural causal model.
> Indeed, lines 5,6,7,8 of algorithm 2 represent the most general version of this idea/trick, and forms an important basis of our overall identification strategy.  No analogue of these lines exist in prior work on data fusion.

---

### Official Review · Reviewer_2Y3A · 2024-03-22

**Q2-1 Originality-Novelty:** 4
**Q2-2 Correctness-Technical Quality:** 3
**Q2-5 Clarity Of Writing:** 4

**Q1 Summary And Contributions:**

The paper considers identifying causal effects from multiple data sets, some of which may suffer from systematic selection bias. They introduce SCMs with selection variables that can be connected in arbitrary ways with other variables, and introduce a novel graphical model (CS-SCM) that can model systematic selection data.
The paper also introduces a hierarchy of selection mechanisms, similar to missing data mechanisms, and give identifiability algorithms for all three settings.

**Q2-3 Extent To Which Claims Are Supported By Evidence:**

4: Excellent: all claims are supported by very convincing evidence (in the form of comprehensive experimental evaluation, rigorous mathematical proofs, detailed (pseudo-)code, precise references, well-motivated and realistic assumptions) and the authors deliver what they promise.

**Q2-4 Reproducibility:**

4: Excellent: key resources (e.g. proofs, code, data) are available and key details (e.g. proof sketches, experimental setup) are comprehensively described for competent researchers to confidently and easily reproduce the main results.

**Q3 Main Strengths:**

-The paper presents a novel approach to modelling systematic selection, defines a new hierarchy of selection (CAR, AR, NAR) and presents novel identifiability results for all three.
-The authors reformulate existing identification algorithms very elegantly using Markov kernels.
-The paper is very well written even though it is quite dense.

**Q4 Main Weakness:**

-I did not find major weaknesses, see detailed comments below.

**Q5 Detailed Comments To The Authors:**

-	Page 3, column 2, definition of $q_{\vec{V} \backslash\{V\}}$ should bbe (\vec{V} \backslash\{V\} \mid \vec{W} \cup\{V\})$ instead of $q_{\vec{V} \backslash\{V\}}(\vec{V} \backslash\{V\} \mid \vec{W} \cup\{W\})$?

-	Page 4, “Since domain … which is often how units from a single superpopulation are assigned to different experimental and observational settings in practice.” I am not sure I understand this sentence. Aren’t selection diagrams treating S’s like a random variable, just one that is not allowed to have no parents (or children, depending on the approach?

-	Related, on Page 5, the authors mention that the main differrence with selectionn diagrams is that S’s are allowed to be causally influenced by other variables in the system, so I am not sure why they are not treated like RVs.

-	It was not completely clear to me which of the reformulations (of ID and gID) are first presented in this paper.

-	Some details of the algorihtms are not discussed at all, for example the complexity or how the reachability condition is checked.

**Q9 Complying With Reviewing Instructions:**

Yes

---

> ### Author Rebuttal · Authors · 2024-04-05
>
> We thank the reviewer for their positive comments. The reviewer raised a number of points which we will address below.
>
> The first point is indeed a typo which we will address, thanks for noticing this.
>
> Regarding the second and third points - selectors have appeared in a variety of contexts. The selectors that have appeared in the works of Bareinboim, Tian, and Pearl are not treated as random variables. In particular, in the proof of theorem 1 of [5], the structural causal models featured in the proof don’t specify a structural equation for S, nor are distributions featuring the selector S before the conditioning bar ever discussed. In addition, the selectors that appear in Phillip Dawid’s formalism of decision theoretic causality are also not random variables. One advantage of our work is that we admit the selector as a first-class random variable that is a part of the structural causal model.
>
> In the notation of our paper, selectors in this prior work are treated as Ws in conditional ADMGs – vertices in the graph corresponding to W in q(V | W) where a joint distribution on V and W may not necessarily exist.  In other words, Ws are "variables" in the sense that they have values w which index distributions q(V | W=w), but W are not random variables themselves! In fact, Dawid introduced a generalization of conditional independence theory that may apply to objects like q(V | W) without requiring a joint on V and W, specifically because his theory of causality with selectors required this!
>
>
> Regarding the fourth point, the ID reformulation is not novel. This can be found at [2].  The reformulation of gID is relatively straightforward given the reformulation of ID, and the original statement of the gID algorithm.
>
> Regarding the last point, we can provide a description of the time and space complexity of the algorithm. As with all identification algorithms in graphical models, we may ask two distinct computational complexity questions.  One would treat the algorithm as a decision procedure answering a YES/NO question about identification of a given query in a given model (and potentially giving additional useful information, such as the identifying functional if the answer is YES).  The computational complexity of this version of our algorithm has an upper bound of a low order polynomial in the number of edges |E| and vertices |V| of the input graph.  Specifically, here are the computational complexity calculations of a number of operations that appear in the algorithm:
> the computation of the set Y* is O(|E| + |V|), for which a depth-first traversal of the graph can calculate this set.
> the computation of districts in Y* is similarly O(|E| + |V|), via a depth first traversal.
> positivity (“laidbackness”) checks are linear in |S|, the size of the state space of S.
> finding a fixable vertex is at most quadratic in |V|, and may be linear with clever use of hashing (since we must intersect descendants and districts).
> Iteration of the fixing operation happens at most O(|V|) times.
> Calculations of reachability of the set entails iterative fixing.
> The overall algorithm is thus at most cubic in |V| + |E| + |S|, and more efficient implementations are likely possible.
> The second computational complexity question is the time it takes to evaluate the identification query itself given, for example, categorical data.  This version of the algorithm is likely intractable for the simple reason that the sum over Y* \ Y in front of the algorithm may be difficult to evaluate in general graphs, for much the same reason that variable elimination and belief propagation algorithms are intractable in dense graphs.  While some prior work exists on making this version of the algorithm tractable using ideas from variable elimination algorithms [1], this work only applies to certain types of sparse graphs, and has not yet been generalized to the algorithm we present (although this is a very interesting area of future work).

---

### Official Review · Reviewer_ib2u · 2024-03-25

**Q2-1 Originality-Novelty:** 3
**Q2-2 Correctness-Technical Quality:** 2
**Q2-5 Clarity Of Writing:** 2

**Q1 Summary And Contributions:**

The paper attempts to provide a general identification algorithm for causal effects in the presence of systematic selections. In particular, the paper considers the setting where there are multiple data sets w.r.t. a same superpopulation, and the task is to see if the desired causal effect can be estimated from the collection of data sets. The paper proposes to consider Context Selected SCMs (CS-SCMs) and the associated Labelled Selection DAGs (LS DAGs), and presents algorithms with theoretical results (soundness and completeness of identification).

**Q2-3 Extent To Which Claims Are Supported By Evidence:**

2: Fair: the main claims are somewhat supported by evidence (but the experimental evaluation may be weak, or does not match entirely with the claims, important baselines may be missing, proofs contain important ideas but lack rigor, algorithmic details are only discussed superficially, references are imprecise, assumptions are not sufficiently motivated or explicated, etc.).

**Q2-4 Reproducibility:**

3: Good: key resources (e.g. proofs, code, data) are available and key details (e.g. proofs, experimental setup) are sufficiently well-described for competent researchers to confidently reproduce the main results.

**Q3 Main Strengths:**

The strength of the paper comes from the attempt to unify analyses w.r.t. causal inference in the presence of selection mechanisms. The paper proposes a general identification algorithms for causal effects, and also provides analyses on soundness and completeness of the algorithm.

**Q4 Main Weakness:**

Overall, the paper is hard to follow. Potential concerns/questions include:

- the claim gets confusing from time to time,
- the framework involves graph objects that differ from each other, and
- the goal of providing a general identification algorithm seems strange (if without identifiability analyses).

The questions are detailed below.

**Q5 Detailed Comments To The Authors:**

- **the claim gets confusing from time to time**

The claim in the paper can get confusing from time to time. For instance, in Section 1, it is mentioned that "units are assigned to different data sets systematically". Here, I assume "units" means individuals. Then, it is mentioned that in missing data literature, "variables may be missing completely at random (MCAR), missing at random (MAR), or missing not at random (MNAR)." It is really confusing. On the one hand, if we are looking at variables, then it is true that certain record/unit may have missing value for a variable. On the other hand, MCAR, MAR, MNAR boil down to different scenarios where (conditional) exchangeability may or may not hold true. How is this related to the student data sets example, where selection is at the unit level instead of that of variables or potential outcomes?

- **the framework involves graph objects that differ from each other**

In the paper, different graph objects are involved, e.g., (LS-)DAGs, SWIGs, ADMGs. These graphs have different properties and not all of them are identical to each. For instance, an edge in a DAG and a MAG does not necessarily encode the same causal semantic. How does this factor into the overall project of providing a general causal effect identification algorithm? This, together with the heavy notation in Algorithms 1 and 2, makes it very difficult to parse the proposal.

- **the goal of providing a general identification algorithm seems strange (if without identifiability analyses)**

The goal of providing a general identification algorithm seems strange. While I understand that Theorem 4 and Theorem 5 attempt to establish soundness and completeness results for the algorithms, shouldn't we first establish unified identifiability conditions for causal effects under the selection hierarchy (if the generality or unification is of interest)? The failure modes listed in Theorem 5 do not seem to correspond to non-identifiability of causal effects, and they only correspond to non-identified causal effect by the algorithm. If this is the case, what is the benefit of considering a general identification algorithm?

**Q9 Complying With Reviewing Instructions:**

Yes

---

> ### Author Rebuttal · Authors · 2024-04-05
>
> We thank the reviewer for their careful reading of our paper.
> We consider the first point raised by the reviewer. We understand how this might be unclear, since by units, we meant ‘experimental units corresponding to rows in our data,’ while by ‘variables being missing’, we meant ‘columns in our data may have ? or NA entries.’ In Section 1 we outlined a possibility where data fusion problems might encounter non-ignorable selection, whereby different individuals (units) are assigned to different datasets not randomly, but by some process, observable or otherwise. We then notice that this situationis similar to that considered in the missingness literature, and by analogy consider a selection hierarchy inspired by the well known missingness hierarchy. The analogy we are aiming to draw between selected data problems we consider and missing data problems is as follows.  In missing data, often individual variables in a row may be censored, however we can imagine a case where the entire row is either fully observed or missing.  In this case we can refer to that row being missing (either completely at random, at random, or not at random).  Similarly, in our case rather than observed or missing, we can imagine that unit is selected into a particular context (observational or interventional), for example to a particular study.  Similarly to missing data, this selection process may be due to an independent coin flip (selected completely at random), a coin flip that depends on observed data shared by all contexts (selected at random), or an arbitrary process that is neither of the previous two (selected not at random).  In addition, just as only some variables in missing data may be censored and others are fully observed, we can imagine that some variables of units may be shared by all contexts (and thus not selected, systematically or otherwise), and others are selected into contexts.  For example, we may measure a vector of baseline covariates on a set of patients and ask them if they wish to enroll in an RCT for a novel treatment.  Individuals who consent to do so will have their subsequent variables measured in an interventional context, while those who do not will have their subsequent variables measured in an observational context corresponding to the usual standard of care.  If the consent choice depends only on baseline covariates, we are in the situation where baseline covariates are not selected, and variables after the consent choice are selected at random.
>
> We consider the second point raised by the reviewer. In ordinary causal inference with graphical models (for example in the way Judea Pearl describes it in his book [7]), 3 types of graphs are also involved.  These include the causal diagram itself (representing a DAG with potentially hidden variables); the latent projection mixed graph (ADMG), first described by [2], which consists of directed (->) and bidirected (<->) edges and which serves as a concise representation of a hidden variable DAG without the need for drawing hidden variables directly; and the mutilated graphs (such as those that arise in the description of the do-calculus rules) which represent interventional contexts.  SWIGs [3] are (arguably) a cleaner version of mutilated graphs with counterfactual variables clearly labeled. A long line of identification work based on these graphis have been considered - see [4, 5] for examples.
> Thus, in our view, our approach is a direct generalization of standard causal inference to handle systematic selection, and has analogues to graphs normal graphical causal inference uses.  Indeed, the only notational innovation in our graphs is the special selection indicator variable S, and edge labels.  In fact, since we treat the selector as a random variable, in some sense our representation is simpler than similar proposals by Dawid, Pearl, Barenboim and others, which view selectors as “indicators of intervention/structural equation change” which are not themselves random variables.  The labels in our graphs simply serve to indicate context specific independence encoding the usual semantics of interventions, whereby a variable that’s intervened to a value ignores the values of parents.
> Conditional ADMGs (CADMGs) which make an appearance in our work have analogues in prior work as well, and correspond to subgraphs serving as inputs to the recursive formulations of the ID algorithm in the works of Tian, Shpitser, Pearl, Huang, Valtorta and others.

---

### Meta-Review · Area_Chair_zXek · 2024-04-19

The paper proposes a sound and complete identification algorithm for multiple environments that might have different forms of systematic selection, extending [Lee et al. 2022, Kivva et al. 2022] to the case of systematic selection. An additional contribution of the paper is to draw an analogy between the hierarchy of missing data problems (MCAR, MAR, MNAR) and a novel hierarchy for systematic selection (SCAR, SAR, SNAR). Another contribution is to propose a joint SCM of the multiple datasets with potentially different selected subpopulations by considering auxiliary selector variables, resembling previous work on selection diagrams, context variables, transportability, and other approaches modelling multiple (interventional) distributions in a single SCM. In this context one of the novelties is that the selector variables can be causally influenced by the other variables in the system.

The reviewers had very split opinions on this paper, as well as a very lively discussion. I will try to summarize the main weaknesses mentioned in the reviews and the discussions, as well as my opinion regarding each point.

**Clarity.** The paper is very technical and quite dense (as also noted by Reviewer 2Y3A) and I can definitely agree with Reviewer ib2u it might be difficult to follow or unclear, especially for a beginner audience.

**Relation between missing data vs systematic selection.** The proposed hierarchy of systematic selection is clearly and very openly inspired by the hierarchy of missingness. This has led Reviewer S8NW and Reviewer ib2u to question the novelty of the algorithm and the applicability of existing work in missigness in this setting, which I think it’s a very fair question and something that the authors should have paid more attention in clarifying in the original manuscript. On the other hand, as the authors mention in the rebuttal, this is a quite different setting, since in one case we might have some values missing for certain columns, while selection would require a whole row to be missing. The modelling and the methods are hence quite different.

**Relation between methods that jointly model interventional distributions and systematic selection.** Reviewer S8NW had also concerns regarding the relation to methods that jointly model different interventions, distribution shifts etc. and the joint modelling of systematic selection. The author rebuttal mentions that in the cited works these “auxilliary” variables are not considered random variables, which I think is a bit missing the point (e.g. they are random variables also for joint SCMs used in some causal discovery methods, e.g. Joint Causal Inference [Mooij et al 2020]). I think the most interesting point is that these selector variables can be also caused by other variables, which to the best of my knowledge is not a very common setting.


**Lack of experimental evaluation.** Reviewer S8NW mentioned this also as a pitfall of the paper. Personally I don’t think this is an issue for this paper since it already provides strong theoretical results and the paper is already quite dense. Additionally, UAI is generally very accepting of such papers.

Overall, taking all of these issues and the other comments into account, I think the strengths of the paper outweigh its weaknesses. I do strongly encourage the authors to make the paper more accessible, possibly also by adding examples in the appendix, or by improving the clarity of the paper. Even as an expert I had to reverse engineer a few things. This hinders the potential impact of the paper, as does the lack of clarity regarding related settings (e.g. missingness).